# Omicron COVID-19 immune correlates analysis of a third dose of mRNA-1273 in the COVE trial

Bo Zhang [1], Youyi Fong[1,2], Jonathan Fintzi[3], Eric Chu[4], Holly E. Janes[1,2], Avi Kenny[5], Marco Carone [5], David Benkeser [6], Lars W. P. van der Laan[7], Weiping Deng[8], Honghong Zhou[8], Xiaowei Wang[8], Yiwen Lu[1], Chenchen Yu[1], Bhavesh Borate[1], Haiyan Chen[9], Isabel Reeder [9], Lindsay N. Carpp [1], Christopher R. Houchens[9], Karen Martins[9], Lakshmi Jayashankar[9], Chuong Huynh[9], Carl J. Fichtenbaum [10], Spyros Kalams [11], Cynthia L. Gay [12], Michele P. Andrasik[1], James G. Kublin[1], Lawrence Corey [1,13], Kathleen M. Neuzil [14,18], Frances Priddy[8], Rituparna Das[8], Bethany Girard[8], Hana M. El Sahly [15], Lindsey R. Baden [16], Thomas Jones[9], Ruben O. Donis [9], Richard A. Koup[17], Peter B. Gilbert [1,2,5] & Dean Follmann [3] ✉, On behalf of the United States Government (USG) COVID-19 Immune Assays Team*, Moderna, Inc. Team*, Coronavirus Vaccine Prevention Network (CoVPN)/Coronavirus Efficacy (COVE) Team*, USG/CoVPN Biostatistics Team*

In the phase 3 Coronavirus Efficacy (COVE) trial (NCT04470427), post-dose two Ancestral Spike-specific binding (bAb) and neutralizing (nAb) antibodies were shown to be correlates of risk (CoR) and of protection against Ancestral-lineage COVID-19 in SARS-CoV-2 naive participants. In the SARS-CoV-2 Omicron era, Omicron subvariants with varying degrees of immune escape now dominate, seropositivity rates are high, and booster doses are administered, raising questions on whether and how these developments affect the bAb and nAb correlates. To address these questions, we assess post-boost BA.1 Spike-specific bAbs and nAbs as CoRs and as correlates of booster efficacy in COVE. For naive individuals, bAbs and nAbs inversely correlate with Omicron COVID-19: hazard ratios (HR) per 10-fold marker increase (95% confidence interval) are 0.16 (0.03, 0.79) and 0.31 (0.10, 0.96), respectively. In non-naive individuals the analogous results are similar: 0.15 (0.04, 0.63) and 0.28 (0.07, 1.08). For naive individuals, three vs two-dose booster efficacy correlates with predicted nAb titer at exposure, with estimates -8% (-126%, 48%), 50% (25%, 67%), and 74% (49%, 87%), at 56, 251, and 891 Arbitrary Units/ml. These results support the continued use of antibody as a surrogate endpoint.

The COVE trial[1,2] (NCT04470427) was a randomized, placebo-controlled phase 3 trial of the mRNA-1273 vaccine. Estimated vaccine efficacy (VE) against COVID-19 in baseline SARS-CoV-2 negative participants was 93.2% [95% confidence interval (CI): 91.0%, 94.8%] over ~5 months of follow-up in the per-protocol analysis. We previously assessed serum IgG binding antibodies (bAbs) against Spike and neutralizing antibodies (nAbs), measured 28 days after dose 2, as correlates of risk (CoRs) and protection (CoPs) against COVID-19 through

A full list of affiliations appears at the end of the paper. *Lists of authors and their affiliations appear at the end of the paper.
✉e-mail: dfollmann@niaid.nih.gov

~4 months post dose 2[3–6]. Using antibody decay models we also demonstrated that the predicted nAb titer at the time of exposure correlated with COVID-19[7]. In all these analyses, the markers were measured against Ancestral Spike and correlated with COVID-19 caused by predominantly Ancestral SARS-CoV-2 lineages[8]. Each marker was shown to be a CoR and CoP of COVID-19, with strongest evidence for nAb markers measured by a pseudovirus (vs. live virus) neutralization assay[3]. Based on these and other analyses, nAb titer has been used as a surrogate endpoint for regulatory authorization or approval of booster doses and variant vaccines[9].

In late 2021, the Omicron variant spread rapidly with successive waves of Omicron subvariants dominating globally by mid-2022[10,11]. All Omicron subvariants have demonstrated some level of immune escape, especially from neutralization by antibodies elicited by natural SARS-CoV-2 infection and/or COVID-19 vaccination[12–19]. The main motivation of this article is to systematically study correlates for Omicron BA.1 and assess whether CoP relationships are applicable across new variants. We studied four markers (i) measured on the day of the booster dose (BD1, "boost"), (ii) measured 28 days later (BD29, "peak"), (iii) fold-rise from BD1 to BD29 ("fold-rise"), and (iv) predicted at the time of exposure (COVID-19 illness visit for cases).

Four objectives were assessed. Objective 1 compared the four BA.1 strain markers with the analogous Ancestral strain markers to assess the importance of variant-matching. Objective 2 compared the BA.1 antibody/Omicron COVID-19 outcome relationship with the Ancestral antibody/Ancestral COVID-19 outcome relationship to check if similar antibody levels are associated with similar reductions in COVID-19 risk. This question was previously investigated by Cromer et al.[20] using a meta-analysis of 15 studies covering Ancestral, Delta, and Omicron waves and demonstrated a strong correlation between estimated nAb titer and vaccine effectiveness against COVID-19. Here we investigate the question through individual-breakthrough analysis of the COVE trial. Previous COVE correlates analyses[3–6] were conducted in baseline SARS-CoV-2 negative (naive) participants. Now most people are non-naive[21,22] and CoPs in non-naive participants might differ from those in naive participants. Objective 3 thus evaluated and compared correlates in naive vs non-naive participants. Objective 4 considered all baseline factors and all markers jointly and used multivariable statistical learning to best predict Omicron COVID-19.

## Results

### Trial schema and participant demographics

A schematic of the COVE trial is provided in Supplementary Fig. 1. Participants randomized to placebo received mRNA-1273 after it was shown to be efficacious and later all participants were offered a booster dose. Blood was drawn on BD1, BD29 and for illness cases at the onset of symptoms (DD1). Assessment of Omicron COVID-19 occurred from December 1, 2021 through April 5, 2022.

We measured antibody from a subset of the entire boosted cohort, namely, COVE participants in the primary series per-protocol cohort (SARS-CoV-2 negative prior to receipt of two doses of mRNA-1273 with no major protocol deviations) who received a third dose of mRNA-1273 vaccine prior to December 31, 2021 ($n = 15{,}713$). Supplementary Fig. 2 illustrates participant flow from enrollment through to the sampling population for the correlates analysis; placebo arm participants with SARS-CoV-2 infection prior to mRNA-1273 vaccination were excluded. Additionally, Supplementary Fig. 2 shows the per-protocol boosted cohort for this study ($n = 14{,}251$) and, via stratified case-control sampling (see Methods), the per-protocol three-dose correlates cohort for whom BD1, BD29 ($n = 218$), and DD1 ($n = 55$) antibodies were measured (Supplementary Table 1 shows the numbers of participants in the per-protocol three-dose correlates cohort across the strata). Participants who tested SARS-CoV-2 RT-PCR+ at their BD1 visit were excluded. In the per-protocol boosted cohort, a study participant was considered SARS-CoV-2 naive (hereafter, "naive") by BD1 if

there was no evidence of SARS-CoV-2 infection (RT-PCR+, Roche Elecsys seropositive, or a symptomatic COVID-19 endpoint followed by positive confirmatory testing) from enrollment to BD1 (including the BD1 visit). A study participant was considered non-naive if the participant showed evidence of infection between 14 days post-dose 2 in the primary mRNA-1273 two-dose series and the BD1 visit (including testing seropositive at BD1; testing RT-PCR+ at BD1 were excluded from the per-protocol boosted cohort as discussed above). Based on this definition, 204 participants were classified as non-naive in the per-protocol boosted cohort and the other 14,047 participants were classified as SARS-CoV-2 naive.

Demographic and clinical information for the per-protocol boosted cohort and the three-dose correlates cohort subset are provided in Supplementary Table 2. Compared to the per-protocol correlates analysis cohort for the blinded-phase COVE correlates analyses[3,4], the per-protocol boosted cohort was similar in age and sex, but lower in baseline risk (24% vs. 40%).

To assess the relative efficacy of the booster dose (three-dose vs. two-dose) and to evaluate antibody markers as correlates of booster protection, a dynamic, unboosted, nonrandomized control group of 2753 participants was also identified. They were in the baseline-negative per-protocol cohort according to the definition in ref. 4; remained in the study through December 1, 2022; and had not received a booster dose by January 31, 2022 (Supplementary Fig. 3).

### Omicron COVID-19 study endpoint

Correlates analyses were conducted for the first occurrence of acute symptomatic COVID-19 with virologically-confirmed SARS-CoV-2 infection, referred to as COVID-19. This adjudicated COVID-19 endpoint was identical to that in the primary analysis[1,2] and the primary series correlates analyses[3–6] of the COVE trial. For BD29 marker correlates analyses, all endpoints starting 7 days after BD29 through to April 5, 2022 (the data cut-off date of the current analysis) were counted (Supplementary Fig. 1); for correlates analyses involving the unboosted control, any remaining unboosted participants were censored on January 31, 2022. Analyses restricted to BA.1 cases ("Omicron COVID-19"), identified by Spike sequencing of the SARS-CoV-2 lineage causing the case whenever possible; lineages for all COVID-19 cases diagnosed before January 15, 2022 were hard-imputed as BA.1. See the Statistical Analysis Plan for Study of Post Dose 3 and Exposure-Proximal Omicron Antibody as Immune Correlates for Omicron COVID-19 in the P301 COVE Study (SAP), available in the Supplementary Information, and Methods for further details.

A non-case (or control) was defined as a participant who showed no evidence of SARS-CoV-2 infection (neither Elecsys+ nor RT-PCR+) between BD1 and April 5, 2022.

### In naive participants, consistently lower bAb and nAb levels at BD29 in Omicron COVID-19 cases vs. non-cases

BD1 Ancestral nAb responses were quantifiable, and BD1 Spike IgG-Ancestral bAb responses were positive, in virtually all per-protocol boosted recipients (Supplementary Table 4; assay limits in Supplementary Table 3; quantifiable/positive response defined in Supplementary Table 4). Response rates tended to be slightly lower for BA.1 nAbs and Spike IgG-BA.1 bAbs (Table 1; quantifiable/positive response defined in Table 1) compared to the Ancestral markers (Supplementary Table 4). In non-cases, the geometric means (GMs) of the BA.1 and Ancestral markers at BD1 were numerically higher for non-naive participants vs. naive participants (Table 1, Supplementary Table 4). Moreover, BA.1 marker levels at BD1 were numerically higher in cases vs. non-cases among non-naive participants and were closer to 1 among naive participants (Table 1). Figure 1 displays individual BD1 and BD29 BA.1 marker levels, with Supplementary Fig. 4 showing a post hoc descriptive analysis of BD1 and BD29 BA.1 marker levels separately in males and females.

**Table 1 | BD1, BD29, and Fold-Rise BA.1 strain neutralizing antibody (nAb) and Spike IgG-BA**

| Status[c] | Marker | Measurement | Omicron Cases[a] | | | Non-Cases[b] | | | Comparison | |
|---|---|---|---|---|---|---|---|---|---|---|
| | | | N[d] | Response Rate[e] (95% CI) | GMC or GMT (AU/ml) (95% CI) | N[d] | Response Rate (95% CI)[e] | GMC or GMT (AU/ml) (95% CI) | Response Rate Difference (Omicron Cases-Non-Cases) (95% CI) | Ratio of GM (Omicron Cases/Non-Cases) (95% CI) |
| Naive | BA.1 Strain nAbs | BD1 | 79 | 84.1% (69.1%, 92.6%) | 11.4 (9.1, 14.3) | 84 | 93.0% (84.6%, 96.9%) | 14.6 (12.1, 17.6) | -0.09 (-0.24, 0.03) | 0.78 (0.58, 1.05) |
| Naive | Spike IgG-BA.1 Strain bAbs | BD1 | 79 | 98.3% (88.3%, 99.8%) | 3621 (2621, 5004) | 84 | 100% (100%, 100%) | 3353 (2646, 4248) | -0.017 (-0.12, -0.002) | 1.08 (0.72, 1.61) |
| Naive | BA.1 Strain nAbs | BD29 | 79 | 100% (100%, 100%) | 259 (194, 347) | 84 | 100% (100%, 100%) | 491 (341, 706) | 0 (0, 0) | 0.53 (0.33, 0.84) |
| Naive | Spike IgG-BA.1 Strain bAbs | BD29 | 79 | 100% (100%, 100%) | 113,143 (87,402, 146,464) | 84 | 100% (100%, 0.0%) | 170,731 (137,772, 211,574) | 0 (0, 0) | 0.66 (0.47, 0.93) |
| Naive | BA.1 Strain nAbs | Fold-Rise (BD29/BD1) | 79 | - | 23 (17.5, 29.4) | 84 | - | 34 (24.6, 46.1) | - | 0.67 (0.45, 1.01) |
| Naive | Spike IgG-BA.1 Strain bAbs | Fold-Rise (BD29/BD1) | 79 | - | 31.2 (24.8, 39.3) | 84 | - | 50.9 (42.0, 61.6) | - | 0.61 (0.45, 0.83) |
| Non-Naive | BA.1 Strain nAbs | BD1 | 32 | 91.4% (74.9%, 97.4%) | 28.3 (17.0, 47.1) | 23 | 96.2% (74.0%, 99.6%) | 19.1 (11.7, 31.1) | -0.048 (-0.22, 0.18) | 1.48 (0.73, 3.00) |
| Non-Naive | Spike IgG-BA.1 Strain bAbs | BD1 | 32 | 100% (100%, 100%) | 7513 (4658, 12117) | 23 | 100% (100%, 100%) | 4406 (2907, 6678) | 0 (0, 0) | 1.71 (0.90, 3.21) |
| Non-Naive | BA.1 Strain nAbs | BD29 | 32 | 100% (100%, 100%) | 346 (231, 517) | 23 | 100% (100%, 100%) | 572 (345, 949) | 0 (0, 0) | 0.60 (0.32, 1.15) |
| Non-Naive | Spike IgG-BA.1 Strain bAbs | BD29 | 32 | 100% (100%, 100%) | 90,534 (63,315, 129,453) | 23 | 100% (100%, 100%) | 148,330 (96,969, 226,897) | 0 (0, 0) | 0.61 (0.35, 1.06) |
| Non-Naive | BA.1 Strain nAbs | Fold-Rise (BD29/BD1) | 32 | - | 12.2 (7.6, 19.7) | 23 | - | 30.0 (18.3, 49.1) | - | 0.41 (0.20, 0.81) |
| Non-Naive | Spike IgG-BA.1 Strain bAbs | Fold-Rise (BD29/BD1) | 32 | - | 12.0 (7.2, 20.0) | 23 | - | 33.7 (22.6, 50.2) | - | 0.36 (0.19, 0.68) |

1 strain binding antibody (bAb) response rates and geometric means by Omicron COVID-19 case vs. non-case status and by SARS-CoV-2 naive vs. non-naive status in the per-protocol boosted cohort, pooled across the original-vaccine and crossover-vaccine arms.

AU/ml arbitrary units/ml, CI confidence interval, GMC geometric mean concentration, GMT geometric mean titer.

[a]Omicron case = COVID-19 Omicron BA.1 endpoint that occurred in the interval [≥7 days post BD29 AND ≥December 1, 2021 to April 5, 2022 data cutoff].

[b]Non-case = No acquirement of COVID-19 (of any strain) in the interval [BD1, April 5, 2022 data cutoff].

[c]Naive = No evidence of SARS-CoV-2 infection from enrollment through to BD1; Non-naive = Any evidence of SARS-CoV-2 infection in the interval [≥14 days after the original two-dose series, BD1].

[d]N is the number of cases sampled into the subcohort within baseline covariate strata.

[e]Definition of "responder" for a marker measured on a given time-point: positive (quantifiable) response defined as BA.1 strain nAb titer on that time-point ≥8 AU/ml; positive response defined as Spike IgG-BA.1 strain bAb concentration on that time-point ≥102 AU/ml.

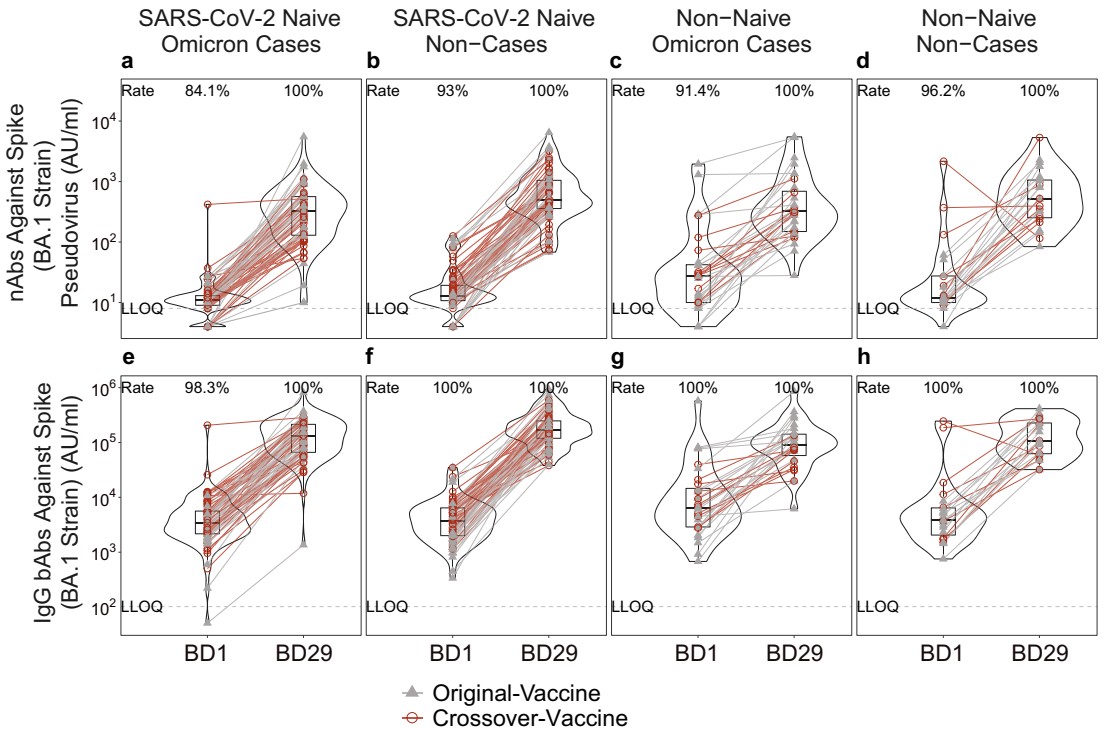

**Fig. 1 | Distributions of BD1 and of BD29 neutralizing antibody (nAb) and binding antibody (bAb) marker levels, stratified by Omicron COVID-19 case vs. non-case status and by SARS-CoV-2 naive vs. non-naive status. a–d** nAb titer against Spike (BA.1 strain) pseudovirus; **e–h** IgG bAb concentration against Spike (BA.1 strain). Points are from per-protocol boosted participants in the original-vaccine (filled triangles) or crossover-vaccine (open circles) arm with lines (gray: original-vaccine arm; red: crossover-vaccine arm) connecting the BD1 and BD29 data points for an individual participant (**a**, **e**: n = 79; **b**, **f**: n = 84; **c**, **g**: 32; **d**, **h**: n = 23). The violin plots contain interior box plots with upper and lower horizontal edges representing the 25th and 75th percentiles of antibody level and middle line representing the 50th percentile. The vertical bars represent the distance from the 25th (or 75th) percentile of antibody level and the minimum (or maximum) antibody level within the 25th (or 75th) percentile of antibody level minus (or plus) 1.5 times the interquartile range. Each side shows a rotated probability density (estimated by a kernel density estimator with a default Gaussian kernel) of the data. Positive response rates were computed with inverse probability of sampling weighting. LLOQ, lower limit of quantification. AU/ml, arbitrary units/ml. LLOQ = 8 AU/ml for nAb BA.1 and 102 AU/ml for Spike IgG BA.1. Positive (quantifiable) response for BA.1 nAb at a given timepoint was defined by value ≥LLOQ at that timepoint. Positive response for Spike IgG-BA.1 bAb at a given timepoint was defined by value ≥LLOQ at that timepoint. Omicron Case = COVID-19 endpoint in the interval [≥7 days post BD29 AND ≥December 1, 2021 to April 5, 2022 (data cutoff date)]. Non-case = Did not acquire COVID-19 (of any strain) in the interval [BD1 to April 5, 2022]. SARS-CoV-2 naive = No evidence of SARS-CoV-2 infection from enrollment through to BD1; Non-naive = Any evidence of SARS-CoV-2 infection in the interval [≥14 days after the first two doses of mRNA-1273, BD1].

At BD29, 100% of correlates subcohort members—both naive participants and non-naive participants—had a quantifiable/positive response for BA.1 nAbs and Spike IgG-BA.1 bAbs (Table 1). GMs at BD29 were lower in Omicron COVID-19 cases vs. non-cases: e.g., in naive participants, the GM ratios were 0.53 (0.33, 0.84) for BA.1 nAbs and 0.66 (0.47, 0.93) for Spike IgG-BA.1 bAbs (Table 1), with similar results for the two Ancestral markers (Supplementary Table 5). Similar GM ratios were generally obtained in non-naive participants (Table 1, Supplementary Table 5), except for Ancestral nAbs where the GM ratio was 0.96 (0.53, 1.73) (Supplementary Table 5).

The lowest GM fold-rise (BD29/BD1) (95% CI) [12.0 (7.2, 20.0)] was observed for Spike IgG-BA.1 bAbs in non-naive cases, with a very similar fold-rise [12.2 (7.6, 19.7)] in BA.1 nAbs in non-naive cases (Table 1). The greatest GM fold-rise [50.9 (42.0, 61.6)] was observed for Spike IgG-BA.1 bAbs in naive non-cases (Table 1). The same pattern was seen for the fold-rise Ancestral markers (Supplementary Table 5).

### Correlations among antibody markers
In naive participants, Spike IgG-Ancestral and Spike IgG-BA.1 bAbs were highly correlated at BD1 (weighted Spearman rank r = 0.90; P < 0.001) and at BD29 (r = 0.96; P < 0.001); Ancestral and BA.1 nAbs were moderately correlated at BD1 (r = 0.63; P < 0.001) and highly correlated at BD29 (r = 0.89; P < 0.001) (Supplementary Figs. 8, 9). Similar results were seen in non-naive participants (Supplementary

Figs. 10, 11). For all 4 markers, BD1 and BD29 levels were weakly correlated among naive participants (e.g., r = 0.38, P < 0.001, for BA.1 nAbs) and among non-naive participants (e.g., r = 0.35, P = 0.80, for BA.1 nAbs) (Supplementary Figs. 12, 13).

### Strong inverse correlations with Omicron COVID-19 risk and BD29 BA.1 markers, as well as bAb fold-rise markers, especially in naive participants
We next assessed the BD1, BD29, and fold-rise markers as CoRs of Omicron COVID-19. First, covariate-adjusted Omicron COVID-19 risk was estimated through 92 days post-dose 3 across a range of marker levels, separately among naive and non-naive participants. As shown in Supplementary Figs. 14 and 15, no evidence of association with COVID-19 was apparent for the BD1 BA.1 or BD1 Ancestral markers, respectively.

In contrast, the two BD29 BA.1 markers each inversely correlated with Omicron COVID-19 in naive participants (Fig. 2a, b) [HR per 10-fold increase (95% CI): 0.31 (0.10, 0.96), P = 0.042 for BA.1 nAbs and 0.16 (0.03, 0.79), P = 0.024 for Spike IgG-BA.1 bAbs; both family-wise error rate adjusted (FWER) P = 0.118] (Fig. 2e). Among non-naive participants, BD29 BA.1 nAbs trended toward correlating with Omicron COVID-19 [HR = 0.28 (0.07, 1.08), P = 0.064 and FWER P = 0.382] and BD29 Spike IgG-BA.1 bAbs inversely correlated with Omicron COVID-19 [HR = 0.15 (0.04, 0.63), P = 0.009 and FWER P = 0.158] (Fig. 1c–e).

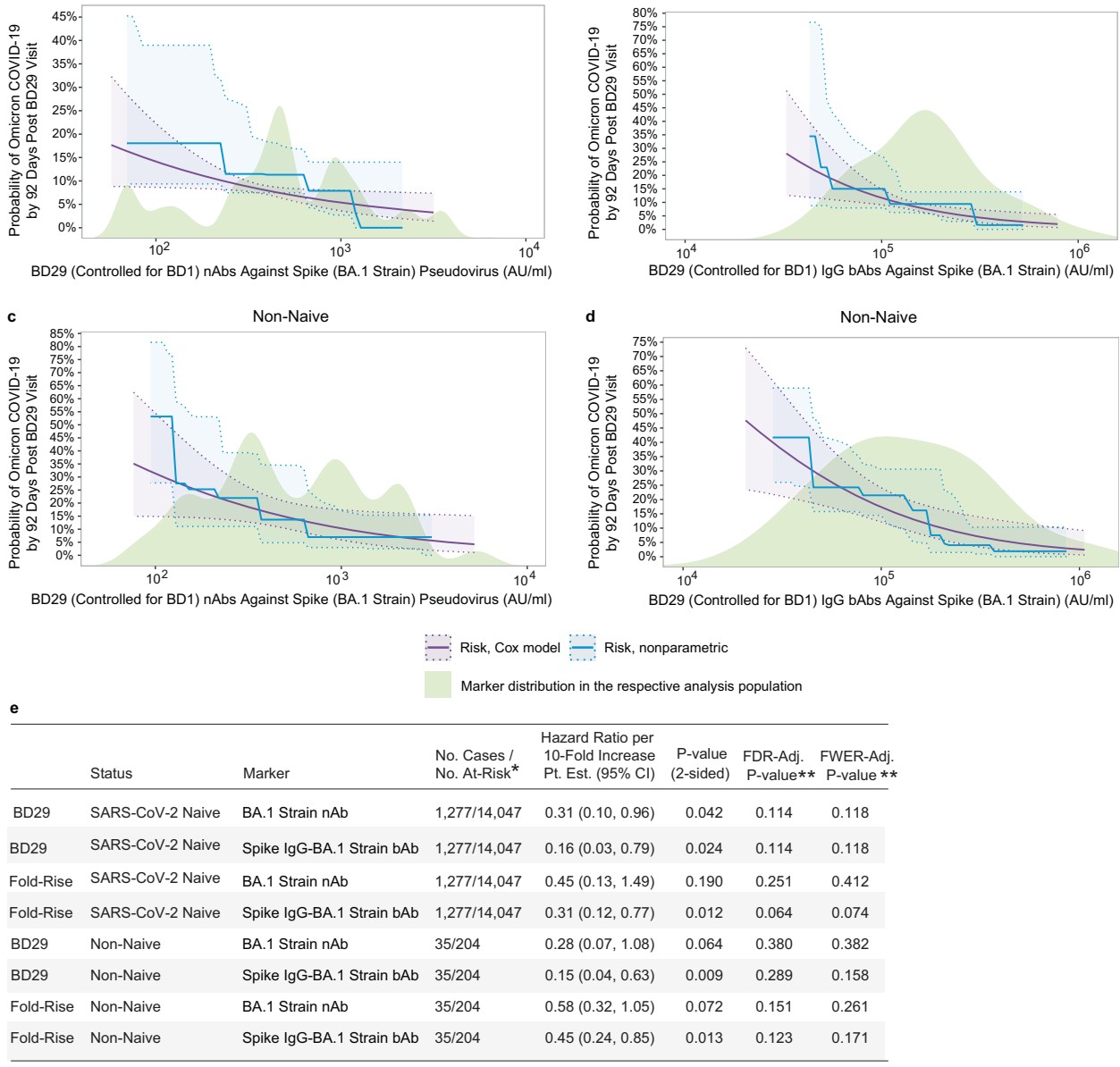

**Fig. 2 | Analyses of BD29 and of Fold-Rise (BD29/BD1) BA.1 strain neutralizing antibody (nAb) titer and Spike IgG-BA.1 strain binding antibody (bAb) concentration as a correlate of risk of Omicron COVID-19.** Curves show cumulative incidence of Omicron COVID-19, estimated using a Cox model (purple) or a nonparametric method (blue), in per-protocol boosted (**a**, **b**) SARS-CoV-2 naive participants (*N* = 14,047) and (**c**, **d**) non-naive participants (*N* = 204) by 92 days post BD29 by BD29 antibody marker level. The solid lines indicate the mean cumulative incidences. The dotted lines and shadings in between indicate bootstrap pointwise 95% CIs. The distribution of the marker in the respective analysis population, calculated by kernel density estimation, is plotted in light green. **e** Hazard ratios of Omicron COVID-19 per 10-fold increase in each BD29 and fold-rise (BD29/BD1) BA.1 marker in per-protocol boosted SARS-CoV-2 naive participants or non-naive participants. Baseline covariates adjusted for: baseline risk score, at risk status, community of color status, BD1 marker level (paired to the BD29 marker studied). *P* values are based on the Wald test and are 2-sided.

Ancestral markers had similar HRs, but with wider CIs (Supplementary Fig. 16). An interaction test was conducted and no evidence of naive/non-naive status modifying the HR was found (interaction *p* = 0.97 for BD29 BA.1 nAb and 0.66 for BD29 Spike IgG-BA.1 bAb).

The bAb fold-rise markers also correlated inversely with Omicron COVID-19 (Fig. 2e; Supplementary Figs. 17 and 18). Among naive participants, the HR per 10-fold increase in fold-rise Spike IgG-BA.1 bAbs was 0.31 (0.12, 0.77), *P* = 0.012 and FWER *P* = 0.074; among non-naive participants, it was 0.45 (0.24, 0.85), *P* = 0.013 and FWER *P* = 0.171. For

fold-rise BA.1 nAbs, the corresponding HRs were 0.45 (0.13, 1.49), *P* = 0.190 and FWER *P* = 0.412 in naive participants and 0.58 (0.32, 1.05), *P* = 0.072 and FWER *P* = 0.261 in non-naive participants (Fig. 2e). Similar results were seen in both populations for the Ancestral fold-rise markers (Supplementary Fig. 18).

An alternative method for assessing markers as CoRs is to divide participants into subgroups defined by antibody marker tertile (Low, Medium, High) and to compare the cumulative incidence curves and hazard ratios across the tertiles. This method differs from the previous

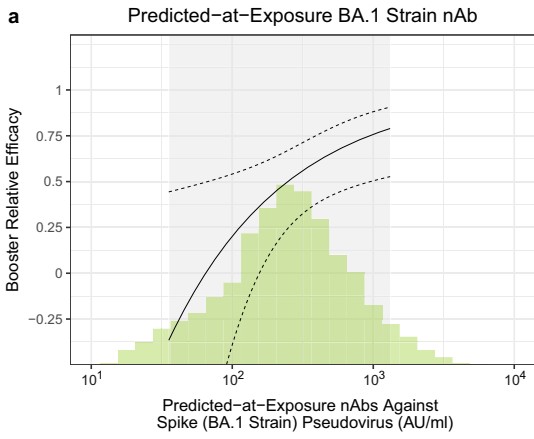
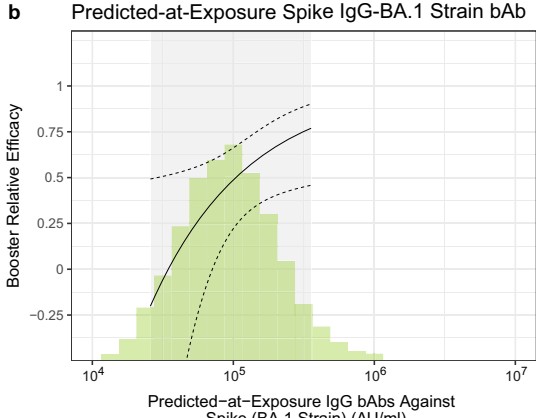

**Fig. 3 | Correlate of booster relative efficacy curves against Omicron COVID-19 among SARS-CoV-2 naive participants (*N* = 2464) as a function of predicted-at-exposure immune marker level. a** Neutralizing antibody (nAb) titer against Spike (BA.1 strain) pseudovirus; (**b**) IgG binding antibody (bAb) concentration against Spike (BA.1 strain). The curves show the relative efficacy of three-dose mRNA-1273 vs. two-dose mRNA-1273. The dashed black lines are 95% confidence intervals. The green histograms are an estimate of the density of predicted-at-exposure antibody marker level in per-protocol boosted SARS-CoV-2 naive participants. The gray shadings indicate the middle 90% (5th percentile to 95th percentile) of this marker distribution.

methods in that it does not rely on any modeling assumptions. Consistent with the analyses described above, no evidence was found to support the BD1 BA.1 or Ancestral markers as inverse correlates of Omicron COVID-19, with HRs (High to Low tertile) generally close to 1 and relatively wide CIs (Supplementary Table 7). In contrast, Cox-model-based marginalized COVID-19 cumulative incidence curves among naive and non-naive participants for subgroups defined by tertile of BD29 BA.1 (Supplementary Fig. 23) and Ancestral (Supplementary Fig. 24) bAbs and nAbs were consistent with inverse correlations. Evidence appeared strongest in naive participants and for the nAb markers, with HR (95% CI) High to Low tertile for BD29 BA.1 nAbs of 0.27 (0.09, 0.78), *P* = 0.016 and FWER *P* = 0.181 and 0.24 (0.07, 0.77), *P* = 0.017 and FWER *P* = 0.101 for BD29 Ancestral nAbs (Supplementary Table 8). These analyses also supported the fold-rise markers as inverse correlates (cumulative incidence curves by tertile shown in Supplementary Figs. 25, 26), with HRs (High vs. Low tertile) indicating inverse correlations and passing FWER-correction for being a significant correlate including Ancestral nAb fold rise [HR = 0.15 (0.05, 0.42), *p* < 0.001, FWER *P* = 0.021 for naive participants and 0.14 (0.04, 0.45), *p* < 0.001, FWER *P* = 0.020] for non-naive participants (Supplementary Table 9).

**Predicted-at-exposure and BD29 antibody correlates of booster relative efficacy among SARS-CoV-2 naive participants**

The analyses reported up to this point have considered antibody markers measured at a fixed time-point relatively close to vaccination. Given that antibody responses wane post-vaccination, however, immune responses at the time of exposure may be better correlates for COVID-19 outcomes, especially over longer follow-up periods, compared to early fixed-time-point measurements. Therefore, we analyzed time-varying predicted antibody levels where the daily risk of COVID-19 depends on the predicted antibody level on that day using a Cox model with calendar time index [see Methods and ref. 7] in naive participants. Predicted antibody levels correlated well with the actual antibody readouts on DD1, thus validating the prediction (Supplementary Figs. 31, 32). Results were similar for the bAb markers.

Receipt of a third dose provided a 46% (20%, 64%) reduction in COVID-19 throughout follow-up compared to a dynamic unboosted (two-dose) control group. Figure 3a, b provide correlates of booster efficacy curves in naive participants for various levels of predicted-at-

exposure antibody by contrasting the hazard of boosted participants with a given predicted-at-exposure antibody level with the overall hazard of the unboosted as a reference group. For boosted recipients with predicted-at-exposure BA.1 nAb titers of 56 AU/ml (10th quantile), 251 AU/ml (median) and 891 AU/ml (90th quantile) at exposure, the proportion reduction in COVID-19 risk for booster relative efficacy (three-dose vs. two-dose) is −8% (95% CI −126%, 48%), 50% (95% CI 25%, 67%) and 74% (95% CI 49%, 876%) (Fig. 3a). Booster relative efficacy results for BD29 BA.1 nAb titer (Supplementary Fig. 33a) were similar [booster relative efficacy = −7% (−113%, 46%) for 102 AU/ml (10th quantile); 56% (33%, 72%) for 479 AU/ml (median); 80% (54%, 91%) for 1738 AU/ml (90th quantile)]. Analyses repeated with BD29 BA.1 Spike IgG-BA.1 bAbs (Supplementary Fig. 33b) and with predicted-at-exposure Ancestral nAbs and Spike IgG-Ancestral nAbs (Supplementary Fig. 34a, b) yielded curves with similar shape.

Analogous analyses with an unboosted control for the non-naive participants were not possible due to extreme confounding; specifically, different events tended to define a participant as non-naive in the boosted vs in the unboosted group. Among the boosted, participants generally became non-naive due to asymptomatic infections in Spring 2021, whereas among the unboosted, participants generally became non-naive due to COVID-19 in Fall 2021.

**Comparison to Ancestral strain correlates study**

We next compared the Ancestral antibody/Ancestral COVID-19 VE curve (2-dose vs. placebo) estimated previously in baseline-negative participants in COVE[4] (Fig. 4a) with the BA.1 antibody/Omicron COVID-19 booster efficacy (3-dose vs. 2-dose) curve in SARS-CoV-2 naive participants (Fig. 4b). Since the Ancestral CoP curve used a nAb assay measured in ID50/ml, we concluded an assay concordance study and scaled the BD29 BA.1 nAb from PPD AU/ml units to ID50/ml units. We then used the previously established conversion factor for Ancestral nAbs to further scale BA.1 nAb titers in ID50/ml units to imputed IU50/ml units (see Methods), making the x-axes in Fig. 4a, b directly comparable. Post dose 3 BA.1 nAb titers were lower than post dose 2 Ancestral nAb titers, with only a limited range of overlap in the highest end of post dose 3 BA.1 nAb titers and the lowest end of post dose 2 Ancestral nAb titers (-100 IU50/ml to -300 IU50/ml). Comparison of the two curves within this range of overlap shows that estimated vaccine efficacy (versus placebo) against Ancestral COVID-19

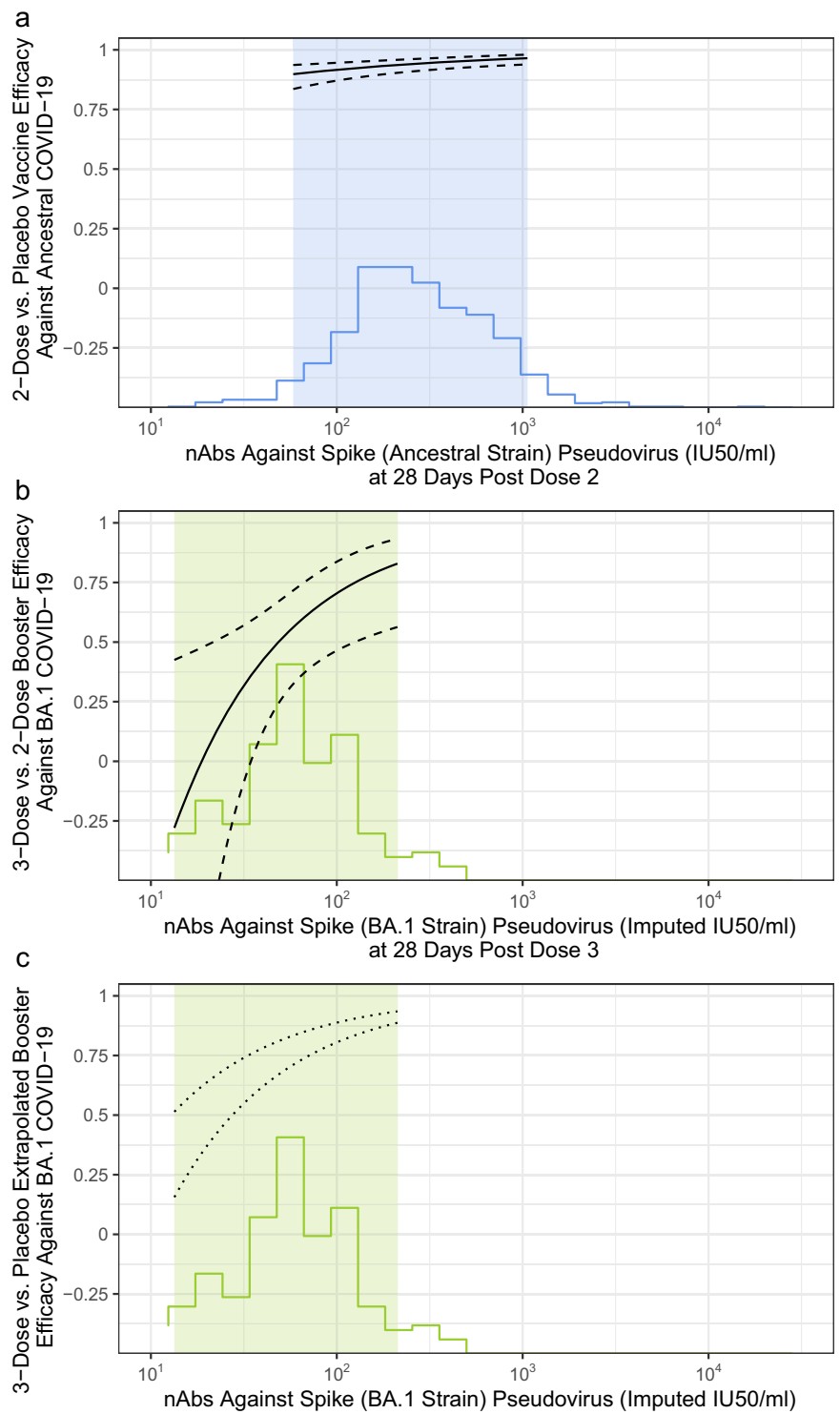

**Fig. 4 | Matched neutralizing antibody, COVID-19 vaccine efficacy curves for Ancestral and Omicron eras.** The curves show vaccine efficacy among SARS-CoV-2 naive participants (**a**, $N = 1615$; **b**, **c**, $N = 2464$). **a** The solid curve graphs two-dose vs. placebo vaccine efficacy against Ancestral COVID-19 by D57 (28 days post dose 2) Ancestral strain neutralizing antibody (nAb) titer in International Units (IU50/ml). The blue histogram shows the distribution of post dose 2 Ancestral nAb titer. The light blue shading indicates the middle 90% (5th percentile to 95th percentile) of the marker distribution. **b** The solid curve graphs three-dose vs. two-dose booster relative efficacy against Omicron COVID-19 by BD29 (28 days post dose 3) BA.1 nAb titer in imputed IU50/ml (see Methods). In (**a**) and (**b**), solid lines are point estimates and dashed lines are 95% confidence intervals. **c** The two dashed lines are the most and least conservative estimates of extrapolated booster vaccine efficacy against Omicron (BA.1) COVID-19 by BD29 (28 days post dose 3) BA.1 nAb titer in imputed IU50/ml for a 3-dose group vs an unvaccinated group. The curves are based on inferring an unvaccinated group using observational cohort data reported in eTable 2 in ref. 23, namely that by 13 months post dose 2, VE (versus an unvaccinated control) against infection and hospitalization waned to 34% and 62%, respectively. In (**b**) and (**c**), the green histogram shows the distribution of post dose 3 BA.1 nAb titer. The light green shadings indicate the middle 90% (5th percentile to 95th percentile) of the marker distribution.

was 91% at post-dose 2 Ancestral nAb titer of 100 IU50/ml (Fig. 4a), whereas estimated vaccine efficacy (3-dose versus 2-dose) against Omicron COVID-19 was 71% at post-dose 3 BA.1 nAb titer of 100 imputed IU50/ml (Fig. 4b).

### Using observational cohort data to attempt to infer an unvaccinated group for comparison to the boosted (three-dose) cohort

The Ancestral CoP curve in Fig. 4a uses an unvaccinated control group while the Omicron curve in Fig. 4b uses a vaccinated (2-dose) but unboosted group. To infer an unvaccinated group for the Omicron curve we reasoned as follows. The two-dose unboosted control group had a median of ~13 months follow-up between dose 2 and December 2021. Observational data have shown that by 13 months post dose 2, VE (versus an unvaccinated control) against infection and hospitalization waned to 34% and 62%, respectively [eTable 2 in ref. 23]. Using these estimates, we scaled the booster efficacy curve of Fig. 4b to derive a CoP curve with an unvaccinated control using Eq. (1):

$$CoP(Ab) = 1 - \{1 - BE(Ab)\} \times (1 - VE) \times 100\% \qquad (1)$$

where BE(Ab) is the Booster Efficacy curve of Fig. 4b and VE is either 0.34 or 0.62, leading to lower and upper bounds for the estimated CoP curve (Fig. 4c). Comparison of the curves in Fig. 4a vs. Fig. 4c in the region of nAb titer overlap (~100–300 IU50/ml) shows slightly lower vaccine efficacy estimates in the Omicron CoP curve (81–89%) vs. the Ancestral CoP curve (91%).

### Predicting Omicron COVID-19 risk

We applied ensemble machine learning to investigate the COVID-19 predictive power of individual or combinations of antibody markers in naive and non-naive boosted participants. Classification accuracy was quantified by the cross-validated area under the receiver operating characteristic (ROC) curve (CV-AUC) and its 95% CI. All models included baseline risk factors (risk score, at risk status, community of color status) (Supplementary Tables 10 and 11). Main results for naive participants are: (1) the best model based on BD1 markers included both nAb and Spike IgG bAb against BA.1 [CV-AUC = 0.641, 95% CI (0.552, 0.721)]; (2) Models that included BD29 markers generally outperformed the BD1-only marker models, where the single-marker model with BD29 nAb against BA.1 achieved CV-AUC = 0.677 (0.591, 0.753); and (3) Models replacing BD29 markers with BD29/BD1 fold-rise markers resulted in lower prediction accuracy [e.g., CV-AUC = 0.582 (0.493, 0.666) for nAb against BA.1]. Main results for non-naive participants are: (4) The best model based on BD1 markers included anti-receptor binding domain (RBD) IgG Ancestral bAbs and nAbs and BA.1 nAbs [CV-AUC = 0.621 (0.465, 0.756)]; (5) Models that included one variant-matching BD29 marker had negligible predictive power given their CV-AUC was lower than the model with baseline risk factors alone [e.g., CV-AUC = 0.559 (0.389, 0.719) for BA.1 nAbs]; (6) Replacing the BD29 bAb marker with its corresponding fold-rise marker improved prediction performance [e.g., CV-AUC = 0.651 (0.496, 0.780) for fold-rise of RBD IgG-Ancestral bAbs vs. 0.577 (0.411, 0.728) for absolute level]; and (7) Multivariable marker models generally performed better than single marker models, with the best model including BD29 Ancestral RBD IgG bAbs and nAbs [CV-AUC = 0.712 (0.558, 0.829)].

## Discussion

In this paper we provide an extensive analysis of the correlation between antibody response to a third dose of mRNA-1273 and Omicron COVID-19 risk during the initial Omicron wave in the COVE trial. Using multiple complementary statistical methods, we evaluated a variety of antibody measurements made at different times to fully evaluate relationships between antibody responses and clinical outcome.

Strengths include use of a randomized clinical trial cohort with rigorous follow-up, careful virologic and symptom sampling, use of validated assays, and standardized analyses that facilitate comparison with the Gilbert et al. correlates analysis[4] following second dose during the Ancestral era of the same trial. We find that antibody measured 28 days post third dose generally correlates with Omicron COVID-19 disease, supporting its continued use as a surrogate endpoint for regulatory decision making. For the BA.1 nAb assay, BD29 titer was estimated to have hazard ratios of 0.31 (95% CI: 0.10,0.96) and 0.28 (0.07, 1.08) for SARS-CoV-2 naive participants and non-naive participants, respectively, similar to the previous estimates for SARS-CoV-2 naive participants with an Ancestral nAb assay, 0.42 (0.27, 0.65). Consistent associations were seen with the bAb assays and with alternate statistical methods of analysis.

Given that the non-naive participants have hybrid immunity, which has been demonstrated to be quantitatively and qualitatively different from vaccination-alone induced immunity[24–28], one of our objectives was to evaluate correlates separately in naive and non-naive participants. In our study, naive and non-naive participants were similar in age (median 52 vs. 54 years, respectively), at-risk status (27% vs. 24%, respectively), and BD29 nAbs (BA.1 nAb GMTs for non-cases: 491 AU/ml vs 572 AU/ml, respectively). We found that the correlates results were very similar between naive and non-naive participants, with e.g., estimated hazard ratios of Omicron COVID-19 of 0.31 and 0.28, respectively, per 10-fold increase in BD29 BA.1 nAb titer. There was also no evidence of naive/non-naive status modifying the hazard ratio, as determined by a formal statistical test. Thus the naive and non-naive groups appear similar in this analysis, i.e., with regard to immune correlates after a third (booster) dose.

In addition to peak BD29 antibody, we evaluated BD1 and BD29/BD1 fold rise as correlates of Omicron COVID-19. BD1 antibody, measured at a median of 11 months post second dose, was poorly correlated with BD29 antibody and generally did not correlate with Omicron COVID-19 In comparison, Hertz et al. measured baseline IgG a median of 6 months from third to fourth Pfizer mRNA vaccine dose and showed individuals with low baseline IgG to index-strain receptor binding domain or index-strain S2 had significantly higher risk of COVID-19 during the Omicron wave of early/mid 2022[29]. BD1 antibodies might have been more predictive had boosting occurred closer to the second dose. Another possible explanation is the difference in the number of doses (booster after 3 doses in Hertz et al. vs. after two doses in the current manuscript). The BD29/BD1 fold rise markers have similar strengths of evidence as correlates as the peak BD29 markers in univariable marker analyses, whereas multivariable analyses suggested weaker evidence in naive participants and stronger evidence in non-naive participants.

We focused on the BA.1 assays as they matched the BA.1 Omicron COVID-19 cases accrued during follow-up. In contrast, the Ancestral bAb and nAb assays were matched to the mRNA-1273 antigen. Unsurprisingly, the BD29 Ancestral bAb and nAb levels were higher than the BA.1 levels for both assays. In naive Omicron COVID-19 cases, for example, the geometric mean BD29 Ancestral nAb titer was 12-fold higher than the geometric mean BA.1 nAb titer (3234 AU/ml vs. 259 AU/ml, Supplementary Table 5 and Table 1). Even so, the BA.1 and Ancestral markers are highly correlated and the hazard ratios in naive participants for BA.1 and Ancestral nAbs are 0.31 and 0.33, respectively and for bAbs are 0.16 and 0.23, respectively; Fig. 2e and Supplementary Fig. 16e). Thus, for SARS-CoV-2 naive participants, variant-antibody matching to variant COVID-19 may not give a meaningfully better prediction of booster relative efficacy. For non-naive participants, correlation point estimates suggested variant-antibody matching improved the correlate, albeit with insufficient precision to draw a conclusion.

Our study allowed a comparison of bAbs and nAbs as correlates. nAb assays appealingly measure in vitro function while bAb assays

have the advantage of less technical measurement error. Measurements from both assays consistently correlated with Ancestral COVID-19 in SARS-CoV-2 naive participants after primary immunization in multiple studies.[9] We demonstrated that BA.1 bAbs significantly correlated with Omicron COVID-19 with very similar hazard ratios and *p*-values for SARS-CoV-2 naive participants and non-naive participants. For BA.1 nAbs, the relationships for naive participants and non-naive participants were again very similar with slightly larger *p*-values than for bAbs (Fig. 2e, Supplementary Fig. 16e). Our results do not demonstrate that one assay readout is a better correlate of Omicron COVID-19 than the other.

An important limitation is the fact that the timing of the boost was not randomized. This could lead to a bias in the relative efficacy of estimates at a given antibody level that compare boosted to unboosted participants (Figs. 3 and 4), although it should have minimal effect on the CoR analyses that compare between antibody levels among boosted participants. The relative efficacy would be overestimated if early-boosted participants tended to be at lower risk; in contrast, the relative efficacy would be underestimated if late/never-boosted participants had lower risk or were less likely to report COVID-19. Although we statistically adjusted for covariates to attempt to address this issue, residual confounding still remains a concern.

Other limitations include: (i) our analysis is for BA.1 Omicron COVID-19 and not current variants; (ii) while we preferentially sampled those with severe COVID-19, it was too rare to study as a separate endpoint; (iii) vaccination was with the Ancestral strain and non-naive participants predominantly acquired Ancestral strain COVID-19; (iv) non-naive participants were largely defined by asymptomatic infection a median of ~8 months prior to boost rather than symptomatic COVID; (v) the sample size for non-naive participants was relatively small such that the multivariable learning analyses in particular had limited precision, and thus these results are interpreted as hypothesis generation in need of additional analysis; and (vi) T-cell and B-cell responses were not studied, such that the contribution of other immune markers as potential correlates of protection could not be assessed.

An important question is the transportability of CoP relationships across new variants. In SARS-CoV-2 naive participants, the HR of Omicron COVID-19 in BA.1 nAb measured at "peak" (BD29) was 0.31 (95% CI: 0.10, 0.96) and relatively similar to the HR of Ancestral COVID-19 in SARS-CoV-2 negative two-dose recipients measured at "peak" post-dose 2 (D57), 0.42 (0.27, 0.65). A different question is whether a given titer from a nAb assay matched to the circulating variant gives the same level of protection across variants—a variant-invariant absolute CoP. In Fig. 4a we showed that a post-dose 2 Ancestral nAb titer of 100 IU50/ml was associated with a 91% reduction in Ancestral COVID-19, compared to placebo, while in Fig. 4c a post-dose 3 BA.1 nAb titer of 100 imputed IU50/ml was associated with between a 81% and 89% reduction in Omicron COVID-19 compared to an extrapolated unvaccinated control. Thus the antibody level required for ~90% protection is similar across variants, a conclusion aligned with the meta-analysis model of ref. 30 who proposed a single vaccine efficacy curve for different variants. While in our study results could be interpreted in IU50/ml units based on two concordance studies (PPD to Duke for BA.1, Duke to IU50/ml for Ancestral D614G), for many correlates studies concordance testing will not be available, in which case convalescent serum scaling is the most effective and practical approach, especially when combining data from different assays and labs.

In summary, we found that neutralizing antibody titer measured 28 days post-third dose generally correlated with Omicron COVID-19, with this conclusion supported by multiple statistical methods of analysis. These findings support the continued use of neutralizing antibody titer as a surrogate endpoint for regulatory decision-making. Moreover, consistent associations were also seen with binding antibody levels.

## Methods

### COVE trial
The COVE trial, conducted in the United States, enrolled adults aged 18 and over at appreciable risk of SARS-CoV-2 infection and/or high risk of severe COVID-19 disease[1,2]. In the primary two-dose series, each mRNA-1273 dose was 100 µg; the third (booster) dose was 50 µg.

### Omicron COVID-19 endpoints
The rationale for excluding COVID-19 endpoints between 1 and 6 days post-BD29 is that participants with these endpoints might had SARS-CoV-2 infection before BD29 and may have generated anamnestic responses that affected the BD29 antibody level. The BD29 study visit was not always 28 days post the BD1 visit because of the allowable study visit windows (within 19 and 45 days, both inclusive, of the BD1 visit; see per-protocol exclusions in Supplementary Fig. 2). COVID-19 cases in COVE were sequenced and we prioritized sampling cases with BA.1 lineage based on sequencing. Of the 79 naive cases, 41 were identified as BA.1 by sequencing, 26 were identified as BA.1.1 by sequencing, and 12 were inferred to be BA.1 based on COVID-19 occurring after January 15, 2022. Of the 32 non-naive cases, 16 were identified as BA.1 by sequencing, 3 were identified as BA.1.1 by sequencing, and 13 were inferred to be BA.1 based on COVID-19 occurring after January 15, 2022.

### Case-control sampling design
The correlates study adopted a case-control sampling design stratified by the COVE trial randomization arm, a participant's SARS-CoV-2 naive and non-naive status at BD1, four calendar periods of BD1 visits, and a person's baseline demographics; see the SAP for details. A total of 218 participants (163 SARS-CoV-2 naive and 55 non-naive) had their BD1 and BD29 antibody markers measured and met the per-protocol criteria to be included in the final analysis (Supplementary Fig. 2). Of these samples, 111 were Omicron COVID-19 cases (79 BD1 SARS-CoV-2 naive and 32 BD1 non-naive) and 107 were non-cases (84 BD1 SARS-CoV-2 naive and 23 BD1 non-naive).

Appendix A of the SAP describes how an Omicron case is approximated by adjudicated COVID-19 case (positive RT-PCR for SARS-CoV-2 with eligible symptoms) ≥ 7 days post BD29 AND ≥ December 1, 2021, given the emergence of Omicron (BA.1) wave. Primary endpoint COVID-19 cases with known Omicron BA.1 lineage were prioritized for sampling.

### Pseudovirus neutralizing antibody assay
Serum nAb activity against SARS-CoV-2 was measured in validated assays utilizing lentiviral vector pseudotyped with full-length Spike of the Ancestral (D614G) strain (referred to as Ancestral) NC_045512.2 (PPD Vaccines VAC62) and with full-length Spike of the BA.1/B.1.529 strain (referred to as BA.1) (PPD Vaccines VAC122). VAC62 and VAC122 utilized the Gen5 Microplate Reader and Imager Software, version 3.08. A four-parameter logistic function was used to fit the reference standard using Statistical Analysis Software (SAS) version 9.2, and the sample antibody concentrations were determined by interpolating the sample responses off the fitted reference standard curve. The readout of each assay is serum antibody concentration Ab[C], reported in units ID50 titer [arbitrary units (AU)/ml] with labeling ID50 (AU/ml). For the Ancestral and BA.1 strains, the lower limit of quantitation (LLOQ) is 10 and 8 AU/ml, respectively; values < LLOQ were set to LLOQ/2. For the Ancestral strain, the upper limit of quantitation (ULOQ) is 281,600 AU/ml; for the BA.1 strain, the ULOQ is 24,503 AU/ml. For each strain, values >ULOQ were assigned ULOQ.

### Binding antibody assay
Serum IgG bAbs against Spike antigens of the Ancestral (D614) (referred to as Ancestral), Gamma, Alpha, Beta, Delta AY4, and Omicron BA.1 strains and against the RBD antigen (Ancestral) were

measured using a validated solid-phase electrochemiluminescence (ECL) S-binding IgG immunoassay (PPD Vaccines VAC123). Software associated with the Meso Scale Discovery (MSD) Plate Reader (MPR) Model No. 600 Discovery Workbench (version 4.0.12.1) was used to generate ECL responses. A four-parameter logistic function was used to fit the reference standard using SAS version 9.2, and the sample antibody concentrations were determined by interpolating the sample responses off the fitted reference standard curve.

The readout of each assay is serum bAb concentration, reported in units AU/ml. A factor to enable conversion from AU/mL to binding antibody units (BAU/ml) was not developed for any of the antigens in the PPD assay, and thus the bAb readouts could not be expressed in international units. The assay limits are listed in Supplementary Table 3. For Ancestral Spike and BA.1 Spike the LLOQ is 69 and 102 AU/ml, respectively. For Ancestral RBD the LLOQ is 79 AU/ml. For each strain, values <LLOQ were set to LLOQ/2. Data analyses restricted to Spike IgG-BA.1 bAbs, Spike IgG-Ancestral bAbs, and RBD IgG-Ancestral bAbs, where the latter marker was only studied in the multivariable statistical learning analyses.

## Statistical methods

All data analyses were prespecified in the SAP. All CoR analyses adjusted for at-risk status [defined in ref. 1], predicted baseline risk score, and community of color classification (all persons other than white non-Hispanic). All controlled correlates of protection analyses further adjusted the BD1 level of the matching antibody marker. All $P$ values for correlations are two-sided.

## Assessment of BD1, BD29, and fold-rise markers as correlates of risk (CoRs)

Univariate analyses assessed each of the BD1, BD29, and Fold-Rise markers as CoRs of Omicron COVID-19 in the per-protocol boosted cohort (original-vaccine and crossover-vaccine arms combined). Analyses were performed as described previously[3,4]. In brief, the survey R package[31] was used to obtain point and 95% confidence interval (CI) estimates of the covariate-adjusted hazard ratio of Omicron COVID-19 across marker tertiles and per 10-fold increase in marker level. The analyses used inverse probability sampling–weighted Cox regression. Wald-based two-sided $P$ values for an association of each antibody marker with Omicron COVID-19 are also reported. The same Cox models were also used to estimate antibody marker conditional cumulative incidence of the COVID-19 primary endpoint, with bootstrap 95% CIs reported. Nonparametric dose-response regression[32] was also used to estimate antibody marker conditional cumulative incidence of the COVID-19 primary endpoint, with influence function–based 95% CIs reported.

Cox regression models were also fitted with an interaction term between BD1 and BD29 marker levels, to investigate whether the BD1 marker modifies the effect of the BD29 biomarker. These models adjusted for at risk status, predicted baseline risk score, and community of color classification, as well as for BD1 marker level.

Point estimates of antibody marker threshold conditional cumulative incidence of Omicron COVID-19 and 95% point-wise CIs were calculated using nonparametric targeted minimum loss–based threshold regression[33].

Machine learning analysis[34] was also performed to estimate the best models for predicting Omicron COVID-19. The multivariable analysis was conducted as before[3], but with a few differences detailed in the SAP.

## Antibody decay and Cox modeling for booster efficacy and exposure proximal correlates

For each participant with antibody measurements on BD1 and DD1, a slope was calculated and the median slope was used to predict antibody concentration or titer at each following day for each individual with BD29 antibody using Eq. (2):

$$Ab_i(d) = Ab29_i + B \times d \tag{2}$$

where $Ab29_i$ is an individual's log10 antibody concentration or titer on BD29, $d$ is the number of days post BD29, and B is the median slope described above. This imputed antibody level was then used as a time-varying covariate in a Cox model shown in Eq. (3):

$$h(t) = h_0(t) \exp\left\{X\alpha + Z(t)\left[\beta_0 + \beta_1 Ab(d(t))\right]\right\} w(t) R(t) \tag{3}$$

where $X$ includes minority status, high risk, and risk score, $t$ is the number of days since 12/1/2021, and $Ab(d(t))$ is an individual's predicted $\log_{10}$ antibody level on day $t$, which is $d(t)$ days post BD29, and $Z(t)$ is 1 after boosting and 0 before, $w(t)$ a weight and $R(t)$ identifies when an individual is in the risk set (i.e., excluded from BD1 to BD29 + 6). The booster relative efficacy (3-dose vs 2-dose) as a function of predicted antibody is given by Eq. (4):

$$1 - \exp(\beta_0 + \beta_1 Ab) \tag{4}$$

where Ab ranges over the distribution of predicted antibody over the course of follow-up.

Booster efficacy estimates remove $Ab(d(t))$ from model (1) when fitting the model and the booster efficacy is given by Eq. (5):

$$1 - \exp(\beta_0) \tag{5}$$

BD29 is also assessed as a correlate using Eq. (3) with $Ab(d(t))$ replaced with BD29 antibody.

Code and a mock data set for these analyses are contained in Supplementary Software 1.

## Comparison to Ancestral strain correlates study

The first blinded phase correlates study[4] estimated how two-dose vs. placebo vaccine efficacy varied by Ancestral nAb titer (Duke D614G pseudovirus neutralization assay) at 4 weeks post dose 2 (Day 57), with Ancestral nAb titer calibrated to the WHO 20/136 International Standard and reported in International Units (IU/ml; conversion of Ancestral nAb titers in 50% inhibitory dilution (ID50) to IU50/ml described in the Supplementary Material and SAP of ref. 4 It is of interest to compare the Ancestral antibody marker, Ancestral COVID-19 vaccine vs. placebo efficacy curve with the BA.1 antibody marker, Omicron COVID-19 booster relative efficacy (three-dose vs. two-dose) curve, to ascertain whether a different amount of variant-matched antibody is needed for high-level booster three dose vs. two dose protection than for high-level two dose vs. placebo protection. To do this, we defined an imputed BA.1 nAb biomarker at BD29 scaled such that it can be absolutely quantitatively interpreted vs. Ancestral nAb in IU/50 ml units. This scaling was accomplished in two steps. First, a PPD/Duke assay concordance study was performed on $n = 250$ samples (results in Supplementary Tables 12 and 13). The results showed that the PPD and Duke assays were highly concordant for both Ancestral and BA.1 nAb titers (Spearman correlation = 0.92 for Ancestral and 0.95 for BA.1). The concordance study also estimated Eq. (5) (see Supplementary Table 14), which describes the relationship between PPD AU/ml and Duke ID50/ml (in the $\log_{10}$-scale) for BA.1 nAb titers:

$$(\text{PPD AU/ml} + 0.303)/1.24 = \text{Duke ID50/ml}. \tag{6}$$

Using this relationship, PPD BA.1 nAb titers in AU/ml were converted to Duke titers in ID50/ml. Second, Duke ID50/ml was converted to IU50/ml using a conversion factor of 0.242 as previously described in ref. 4 Note that the conversion factor of 0.242 for Ancestral nAbs was established based on calibration of Ancestral nAbs to the WHO anti-

SARS CoV-2 Immunoglobulin International Standard (20/136). For BA.1 nAbs, given that we need to make the assumption that the same conversion factor of 0.242 (as for Ancestral nAbs) can be used for BA.1 nAbs to convert from Duke ID50/ml to IU50/ml (as the WHO International Standard for anti-SARS-CoV-2 immunoglobulin has not been assayed against Spike-BA.1 pseudovirus, to enable calibration of Duke BA.1 nAbs in ID50 to IU50/ml), we term the units of the converted BA.1 nAbs "imputed IU50/ml". The BD29 booster relative efficacy curve analysis was repeated for this biomarker, and results overlaid with the original Day 57 vaccine efficacy curve analysis, providing a means for absolute comparison of variant-matched titer levels associated with efficacy.

### Inclusion and ethics

The mRNA-1273-P301 study was conducted in accordance with the International Council for Harmonisation of Technical Requirements for Pharmaceuticals for Human Use, Good Clinical Practice guidelines, and applicable government regulations. The Central Institutional Review Board approved the mRNA-1273-P301 protocol and the consent forms. All participants provided written informed consent before enrollment. Central IRB services for the mRNA-1273-P301 study were provided by Advarra, Inc., 6100 Merriweather Dr., Suite 600, Columbia, MD 21044. All necessary patient/participant consent has been obtained and the appropriate institutional forms have been archived.

Site PIs were invited as co-authors and were given the opportunity for intellectual contribution.

### Reporting summary

Further information on research design is available in the Nature Portfolio Reporting Summary linked to this article.

## Data availability

Access to patient-level data presented in this article and supporting clinical documents by qualified external researchers who provide methodologically sound scientific proposals will be available upon reasonable request. Such requests can be made to Moderna Inc., 200 Technology Square, Cambridge, MA 02139, email: datasharing@modernatx.com. A materials transfer and/or data access agreement with the sponsor will be required for accessing shared data. All other relevant data are presented in the paper. The protocol is available online at ClinicalTrials.gov: NCT04470427.

## Code availability

All analyses were done reproducibly on the basis of publicly available R scripts. A portion of these are hosted on the GitHub collaborative programming platform[35]. The rest of these are contained in the Supplementary Software file.

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

## Acknowledgements

This research was supported in part by the Administration for Strategic Preparedness and Response, Biomedical Advanced Research and Development Authority Contracts No. 75A50120C00034 (P3001 study) and No. 75A50122C00013 (Immune Assays); the National Institute of Allergy and Infectious Diseases (NIAID) of the National Institutes of Health (NIH) through grants R37AI054165 and UM1AI068635 to P.B.G., AI068614 to L.C., and 3UM1AI148575 (Baylor College of Medicine VTEU) to H.M.E.S.; and by the National Cancer Institute of NIH through Contract No. 75N91019D00024. The findings and conclusions herein are those of the authors and do not necessarily represent the views of the Department of Health and Human Services or its components. The content is solely the responsibility of the authors and does not necessarily represent the official views of the National Institutes of Health. The content of this publication does not necessarily reflect the views or policies of the Department of Health and Human Services, nor does mention of trade names, commercial products, or organizations imply endorsement by the U.S. Government.

## Author contributions

B.Z., Y.F., P.B.G., and D.F. conceptualized the study. F.P., R.D., B.G., and X.W. also contributed to study concept and design. B.Z., Y.F., J.F., A.K., M.C., P.B.G, and D.F. developed methodology used in the study. B.Z., Y.F., H.E.J., J.F., A.K., M.C., D.B., W.D., H.Z., X.W., Y.L., C.Y., B.B., H.C., I.R., P.B.G., and D.F. curated the data. B.G., F.P., R.D., and X.W. were involved in data collection. B.Z., Y.F., H.E.J., J.F., E.C., A.K., M.C., L.W.P.v.d.L., W.D., H.Z., X.W., Y.L., C.Y., B.B., H.C., I.R., R.D., P.B.G., and D.F. conducted the analyses. B.Z., Y.F., H.E.J., J.F., A.K., M.C., D.B., W.D., H.Z., X.W., Y.L., C.Y., C.R.H., K.M., L.J., C.H., J.M., C.J.F., S.K., C.L.G., M.P.A., J.G.K., L.C., K.M.N., R.P., H.M.E.S., L.R.B., T.J., R.O.D., R.A.K., and P.B.G. contributed resources to the project. B.Z., Y.F., J.F., A.K., M.C., D.B., L.W.P.v.d.L., P.B.G, and D.F. developed software used in the analyses. C.R.H., C.H., R.O.D., R.A.K., and P.B.G. performed project administration. B.Z., Y.F., P.B.G., and D.F. validated the analysis results. B.Z., L.N.C., P.B.G., and D.F. wrote the original draft. All coauthors reviewed and edited the draft. ICMJE guidelines for authorship have been adhered to.

## Funding

## Competing interests

All authors have completed the ICMJE uniform disclosure form at www.icmje.org/coi_disclosure.pdf and declare: W.D., H.Z., X.W., F.P., R.D., and B.G. are employed by Moderna Inc. and have stock or stock options in Moderna Inc. H.E.J. serves/served as an unpaid member within the past 36 months as a DSMB member for a Phase II monkeypox vaccine study (DMID) and for a Phase I oral cholera vaccine study (DMID). M.C. reports an honorarium for service within the past 36 months on a grant review panel for the National Comprehensive Cancer Network and Pfizer. C.J.F. reports grants to his institution within the past 36 months from Gilead Sciences, ViiV Healthcare, and Merck, as well as payment for advisory board participation within the past 36 months from ViiV Healthcare and Theratechnologies, Inc. K.M.N. reports grants to her institution within the past 36 months from Pfizer to conduct clinical trials of COVID-19 vaccines, but K.M.N. receives no salary support from these grants. K.M.N. reports a grant from Vaxco within the past 36 months for phase 1 testing of a broadly reactive COVID-19 vaccine. F.P. was a member of the CEPI Scientific Advisory Board within the past 36 months and reports support from Moderna, Inc. for attending meetings and/or travel within the past 36 months. Within the past 36 months, L.R.B. was involved in HIV and SARS-CoV-2 vaccine clinical trials conducted in collaboration with the NIH, HIV Vaccine Trials Network (HVTN), Covid Vaccine Prevention Network (CoVPN), International AIDS Vaccine Initiative (IAVI), Crucell/Janssen, Moderna, Military HIV Research Program (MHRP), the Gates Foundation, and Harvard Medical School; as well as participated on a DSMB for the NIH and an AMDAC Committee for the FDA. P.B.G. served as an unpaid member of the Moderna Zika Vaccine Scientific Advisory Board within the past 36 months. All authors declare no other support from any commercial entity for the submitted work; no other financial relationships with any commercial entities that might have an interest in the submitted work in the past 36 months, and no other relationships or activities within the past 36 months that could appear to have influenced the submitted work.

## Additional information

[1]Vaccine and Infectious Disease Division, Fred Hutchinson Cancer Center, Seattle, WA, USA. [2]Public Health Sciences Division, Fred Hutchinson Cancer Center, Seattle, WA, USA. [3]Biostatistics Research Branch, National Institute of Allergy and Infectious Diseases, National Institutes of Health, Bethesda, MD, USA. [4]Clinical Monitoring Research Program Directorate, Frederick National Laboratory for Cancer Research, Frederick, MD, USA. [5]Department of Biostatistics, University of Washington, Seattle, WA, USA. [6]Department of Biostatistics and Bioinformatics, Rollins School of Public Health, Emory University, Atlanta, GA, USA. [7]Department of Statistics, University of Washington, Seattle, WA, USA. [8]Moderna, Inc, Cambridge, MA, USA. [9]Biomedical Advanced Research and Development Authority, Washington, DC, USA. [10]Division of Infectious Diseases, Department of Medicine, University of Cincinnati, Cincinnati, OH, USA. [11]Department of Pathology, Microbiology and Immunology, Vanderbilt University Medical Center, Nashville, TN, USA. [12]Department of Medicine, Division of Infectious Diseases, UNC HIV Cure Center, University of North Carolina at Chapel Hill School of Medicine, Chapel Hill, NC, USA. [13]Department of Laboratory Medicine and Pathology, University of Washington, Seattle, WA, USA. [14]Center for Vaccine Development and Global Health, University of Maryland School of Medicine, Baltimore, MD, USA. [15]Department of Molecular Virology and Microbiology, Baylor College of Medicine, Houston, TX, USA. [16]Brigham and Women's Hospital, Boston, MA, USA. [17]Vaccine Research Center, National Institute of Allergy and Infectious Diseases, National Institutes of Health, Bethesda, MD, USA. [18]Present address: Fogarty International Center, National Institutes of Health, Bethesda, MD, USA. ✉e-mail: dfollmann@niaid.nih.gov

## the United States Government (USG) COVID-19 Immune Assays Team

Christopher R. Houchens[9], Karen Martins[9], Lakshmi Jayashankar[9], Chuong Huynh[9], Ruben O. Donis [9] & Richard A. Koup[17]

## Moderna, Inc. Team

Weiping Deng[8] & Honghong Zhou[8]

## Coronavirus Vaccine Prevention Network (CoVPN)/Coronavirus Efficacy (COVE) Team

Carl J. Fichtenbaum [10], Spyros Kalams [11], Cynthia L. Gay [12], James G. Kublin[1], Lawrence Corey [1,13], Kathleen M. Neuzil [14,18], Hana M. El Sahly [15] & Lindsey R. Baden [16]

## USG/CoVPN Biostatistics Team

Bo Zhang [1], Youyi Fong[1,2], Jonathan Fintzi[3], Eric Chu[4], Holly E. Janes[1,2], Avi Kenny[5], Marco Carone [5], David Benkeser [6], Lars W. P. van der Laan[7], Yiwen Lu[1], Chenchen Yu[1], Bhavesh Borate[1], Lindsay N. Carpp [1], Peter B. Gilbert [1,2,5] & Dean Follmann [3] ✉

A full list of members and their affiliations appears in the Supplementary Information.

