## [Peer Review File · Nature Communications]

Omicron COVID-19 Immune Correlates Analysis of a Third Dose of mRNA-1273 in the COVE TrialReviewers' Comments:

Reviewer #1:

Remarks to the Author:

Positive feedback.

I applaud the authors by taking the existing clinical trial and performing additional analyses on the efficacy of the third vaccination with Moderna COVID19 vaccine; these are important analyses. The authors also provided a lot of materials for review and performed several additional analyses, e.g., machine learning, that is admirable. The main text is short and to the point, and discussion includes some limitations which is a good sign.

Major comments

1. The model used to correlate Ab levels with protection is not well described. The model for Ab decay is presented (but no number for equation) but how it enters the protection is unclear. The relationship between Ab level and protection may not follow a simple exponential function (e.g., see 37507368 with examples of alternative models fitting the data on protection well). Whether results change if there is cooperativity or competition in Abs for protection. This needs to be addressed.
2. Non-randomized nature of individuals with or without third vaccination is worrisome. I think authors need to present good arguments what biases this may have introduced and how their estimates of efficacy may thus be incorrect.
3. This work measures Ab levels in arbitrary units (AU/ml) while previous work on efficacy of 2 dose vaccination used international units (IU/ml). Other studies (e.g., 34002089) used other ways to "standardize" Ab levels. Justification of the process is discussed in supplement but it is critical, and should be moved to main text. Why were the measurements done in AU and not IU advocated for in previous works on COVE trial? Could this use of different metrics skew results in some ways?
4. I found it sad that authors mention a recent study (36964146) but spend little time discussing of how the results in this paper are or are not consistent with those published previously. I also think having a more thorough discussion with results from other studies by the Davenport group could be useful, especially in settings of how Ab levels are "scaled" between different individuals and/or studies.
5. I found the statement about data and code availability to be unacceptable. "Access to participant-level data and supporting clinical documents with qualified external researchers may be available upon request and is subject to review once the trial is complete". First, key authors are from the NIH that requires ALL NIH-supported work to be publicly available via repositories. The statement authors provide is a double standard. Second, the data presented were clearly sufficient (per authors) for conclusions, so these anonymized data MUST be made available. Third and finally, codes for the analysis, include those with machine learning, must be made available too, for

verification purposes, and as a part of reproducible science (e.g., on github).

6. I like that authors list some limitations of their work. However, limitations discussed seems minor and they did not list/discuss important ones. These are critical limitations: non-randomized nature of the study, whether individuals had Omicron or not, kinetics of Ab is not fully known given few time points sampled, is HR linear as stated, etc. Limitations should focus more on the actual data and methods used and how these methods (or collected data) could be wrong or biased. Please improve.

7. I find the definition of naives vs. non-naives to be a bit shaky. Given asymptomatic infections the way to determine non-naives would be to do serology with Ags that are not in the vaccine (e.g., N protein). Can you better clarify the division into naives vs. non-naives and caveats associated with such division? Perhaps showing results when you do not divide volunteers into these subgroups could be useful.

Minor comments

I found many figures especially in supplement being of a somewhat poor quality. One poor feature is different ranges in panels of the same graph, e.g., Figure 2 has all different scales, numbers are barely visible; same for Fig 3 and several figures in supplement. To help interpret the data, having the same range for similar measurements (e.g., Ab titer) is important. I was not able to interpret 4 different styles of symbols and 2 colors in Fig 1 - perhaps use gray and red. Not sure about cases vs. non-cases division - these are hard to judge visually. Perhaps use different panels for the two groups?

Figure S5 - what are the red lines? Splines? It should be mentioned in the caption (same for S6). Why are there weird peaks in red lines in A and D?

Figure S7 - move the labels from the top to the bottom. We, readers, used to look at axis names on the left and at the bottom. Same for other figures till S12.

In main text, in addition to the 95 CIs, listing p values from the relevant test would be useful.

It seems that results in line 137-148 have multiple comparisons. Did you correct for multiple comparisons here?

Describing Ab loss as a simple exponential curve may be incorrect. How different would your main results on protection be if the decay is bi-phasic as suggested in several recent studies (PMID: 37425815, 38395697) with a faster initial decay ($T_{1/2} \sim 2-3$ wks).

In machine learning, how did you control for overfitting? This needs to be explained.

Line 259: 95CIs include value >1 which is impossible. It seems that the authors did not do CI calculation properly. Please address.

Line 194: how is it possible that using titers against two different strains gave the same results? Isn't that weird? This should be discussed in the Discussion. Is this due a limitation of the work?

209-212 - are these differences statistically different? This is important to discuss in light that similar vaccines do not induce high titer Abs against Omicron.

258-260 - how come naives vs. non-naives have the same HRs? Isn't that weird?

270-272 - why is this conclusion? Some discussion is needed.

273-280 - How do these results relate to other work published on Abs to Omicron and predicted protection? Some reflection would be important.

314-316 - yes, 85-91% is lower than 96% but still surprisingly high. Why? I thought that these vaccines are very poor protectors against COVID19 caused by Omicron.

Reviewer #2:

Remarks to the Author:

This is an thorough and detailed treatment of the question of protection induced by a booster dose of COVID-19 vaccine against Omicron BA1. Findings confirm that both binding and neutralising antibodies to BA1 can continue to be useful to regulators and vaccine developers as correlates of protection, with no indication that one readout is particularly to be preferred. There is therefore a likelihood that this applies to other variants as well.

The paper is a bit hard to read in place - the results section has a lot of acronyms and a shorthand style of writing. This may be due to word limit and an attempt to synthesise the huge number of results presented.

I have no suggestions for specific changes

Minor notes:

Figure S3 has an "XX" in it suggesting maybe something was supposed to be filled in before submission

Reviewer #3:

Remarks to the Author:

Zhang et al. analyze individual-level data on binding and neutralizing antibody titers (bAb and nAb, respectively) to assess the correlation with Omicron COVID-19 in a non-naïve cohort. Their main conclusion is that antibodies remain a useful surrogate endpoint for vaccine effectiveness.

The main strength of this work is the individual-level data from a large randomized controlled trial that allows Zhang et al. to address their objectives. However, the novelty and significance of their work are not made sufficiently clear, the writing lacks clarity, and it is suggested that higher antibody levels are necessary to be protected from Omicron COVID-19 compared to Ancestral COVID-19 which is not well supported by the results in this reviewer's opinion.

This reviewer's recommendation is a major revision to address the major comments below.

Major comments:

1. The novelty and significance are unclear to this reviewer.

Some of the objectives of this work were already investigated using population-based data (as the authors also state in the introduction). What then is the advantage and significance of repeating this analysis using individual-level breakthrough analysis? Are their results supporting previous population-based conclusions or contradicting them?

The manuscript could be improved substantially if the authors stated explicitly the advantages of their approach.

2. Comparison of Ancestral correlates and Omicron correlates.

At the end of the conclusion, the authors suggest that higher titers are required for protection from Omicron COVID-19 compared to Ancestral COVID-19. This reviewer thinks that this is not well supported by the provided evidence. The proposed adjustment between Ancestral and Omicron titers is not clear (see comment below) and the authors state that it is a crude adjustment, it is not investigated whether a potential difference is significant or negligible considering vaccine efficacy (VE) confidence bounds, and the potential confounder of the variant-matching vaccine for Ancestral but not Omicron is not discussed (may there be immune priming that influences results?).

In the paragraph "Comparison to Ancestral strain correlates study", the last sentences suggests that the two curves in Figure 4 are comparable but in this reviewer's opinion they are not. The Ancestral curve shows efficacy compared to placebo (i.e., presumably relatively high risk and low

efficacy) and the Omicron curve shows efficacy compared to 2 vaccine doses (i.e., presumably low risk of COVID-19 and thus high efficacy as the Ancestral vaccine provides protection also against Omicron COVID-19). For this reason, it is to be expected that the Omicron curve is below the Ancestral curve. This is addressed in the following paragraph but should also be highlighted in this section.

Based on the “Using observational cohort data to attempt ...”, it is this reviewer’s understanding that the VE by 13 months post dose 2 of 34% refers to protection from infection and the VE of 62% refers to protection from hospitalization, however, they appear to be used as the lower and upper bound of the VE (for protection from symptomatic infection?) at the end of this paragraph and in the conclusion (last paragraph). Please clarify the interpretation and the calculation.

The authors suggest a method to adjust the Omicron curve to compare it to the Ancestral curve (see comment above). It would be helpful to visualize this adjusted Omicron curve and directly compare it with the Ancestral curve, instead of comparing the VE at selected titers.

3. Clarity and readability.

Please give more relevant information for clarity and readability. For example, in the section “Predicted-at-exposure and BD antibody...”, the second paragraph appears to refer to naïves only – based on the caption of Figure 3 and the following paragraph – but this is not made clear in the paragraph.

Explain terms at their first use in the text. For example, in “Antibody marker response rates and levels” paragraph 1, lines 5-6, “non-cases”, “non-naïves”, and “naïves” appear to be first used but they are not explained.

Providing motivation, introduction, or background as well as a clear theme or topic within sections and paragraphs can improve the readability of the manuscript.

For example, no motivation, introduction, or background are included in the abstract. Result section paragraphs often start with “Figure XX shows ...” and list results without providing a motivation for the analysis or connecting the different sections. Being more selective with which results are presented (and potentially making a summarizing table for other results) and then adding more explanation and a connection to presented results may also improve readability.

Minor comments:

4. The authors group individuals as naïve or non-naïve and Omicron cases or non-cases and analyze, e.g., naïve and non-naïve individuals separately. However, in this reviewer’s opinion they do not sufficiently discuss similarities and differences between these groups. For example, GM fold-rise smaller in non-naïve cases may suggest a smaller fold-change in Ab titers with increasing number of exposures (vaccine or infection). Are there other differences between these groups and what do they mean or how can they be interpreted? It would be beneficial to add a paragraph to the discussion to summarize and interpret the results in this way.

5. Introduction/background paragraph 2, line 4: please replace “To update our work” with a motivation or main question for this work.

6. In this reviewer's opinion it would be very helpful to add a schematic (e.g., one similar to Figure S1 but specifically for the data used in this work) to explain the nomenclature.

7. Please add p-values to the correlations in the paragraph "Correlations among antibody markers"

8. Conclusion, paragraph 2: the authors hypothesize that the timing of boosting affects the predictive power of baseline antibody titers. Is it possible that the number of vaccines also influences this? For example, those with a low antibody titer after 3 vaccine doses may respond less to vaccination and thus be at a higher risk of COVID-19 compared to those with a low antibody titer after 2 vaccinations (as some of them may have higher titers after the third vaccine).

9. Conclusion, paragraph 3: since the authors scaled the Ancestral nAb titers with a multiplicative factor to be comparable to Omicron nAb titers (in Figure 4), it is not surprising that variant-matching does not increase predictive power. The ability to scale titers in this way could be included as further evidence to support their conclusion.

REVIEWER COMMENTS

Reviewer #1 (Remarks to the Author):

Positive feedback.

I applaud the authors by taking the existing clinical trial and performing additional analyses on the efficacy of the third vaccination with Moderna COVID19 vaccine; these are important analyses. The authors also provided a lot of materials for review and performed several additional analyses, e.g., machine learning, that is admirable. The main text is short and to the point, and discussion includes some limitations which is a good sign.

Response: Thank you for the positive feedback.

Major comments

1. The model used to correlate Ab levels with protection is not well described. The model for Ab decay is presented (but no number for equation) but how it enters the protection is unclear. The relationship between Ab level and protection may not follow a simple exponential function(e.g., see 37507368 with examples of alternative models fitting the data on protection well). Whether results change if there is cooperativity or competition in Abs for protection. This needs to be addressed.

Response: We appreciate the opportunity to clarify. We have added an equation number for antibody decay [Equation (2), p. 24] and now include an equation [Equation (4), p. 25] for booster relative efficacy as a function of imputed antibody level. On p. 25 we state:

“The booster relative efficacy (3-dose vs 2-dose) as a function of predicted antibody is given by Equation (4):

$$1 - \exp(\beta_0 + \beta_1 Ab) \quad (4)$$

where Ab ranges over the distribution of predicted antibody over the course of follow-up.”

We have also done additional sensitivity analyses by considering different functions for how Ab enters into Equation (3) (p. 24):

$$h(t) = h_0(t) \exp\{X\alpha + Z(t) [\beta_0 + \beta_1 Ab(d(t))]\}w(t) R(t) \quad (3)$$

These sensitivity analyses are given in the table below.

We see that there is a large increase in both the robust score and generalized Wald statistics when the including Ab(d) term, but little additional improvement when including a quadratic term or when fitting a 2 degree of freedom spline.

Model	Model Equation	Degrees of freedom	Robust Score Test	Generalized Wald Test
Intercept Only	$h_0(d) \exp\{Z[\beta_1] + X\alpha\}$,	4	19.23	13.98
Log-Linear	$h_0(d) \exp\{Z[\beta_0 + \beta_1 Ab(d)] + X\alpha\}$,	5	32.26	21.94
Quadratic	$h_0(d) \exp\{Z[\beta_0 + \beta_1 Ab(d) + \beta_2 Ab(d)^2] + X\alpha\}$,	6	32.96	25.50
Spline	$h_0(d) \exp\{Z[\beta_0 + f\{Ab(d)\}] + X\alpha\}$, where f() is a natural cubic with two degrees of freedom and an interior knot at the median of the distribution of Ab(t) = 2.35	6	33.07	25.21

2. Non-randomized nature of individuals with or without third vaccination is worrisome. I think authors need to present good arguments what biases this may have introduced and how their estimates of efficacy may thus be incorrect.

Response: We agree that the non-randomized nature of receiving a third dose or not is an issue. We included covariates, minority status, high risk, and risk score to try and ameliorate this concern. Still the estimates of booster relative efficacy may be subject to bias. While the extent of this bias is difficult to gauge, the relatively small unboosted group in this study might have an event rate lower that is biased down, thus attenuating the estimates of booster relative efficacy. In another COVID-19 vaccine trial we observed that the small number of placebo participants who did not get the vaccine, when available within the trial, had an implausibly low attack rate.¹ This was also observed in COVE during the spring of 2022.² Perhaps the few unboosted had effectively disengaged from the trial and were less likely to report COVID-19 symptoms. We addressed this by censoring those who remained unboosted on 31 January 2022. Another possibility is that the early boosted were higher risk which would bias the estimate of booster relative efficacy up, though we endeavored to address this concern by the inclusion of the risk score and high risk covariates. Importantly, the relative effect of different levels of antibody, whether at D15 or time-varying, does not suffer from this concern as the unboosted control group is not involved.

We have added a new paragraph to the discussion where we discuss this concern as a limitation of the present work (p. 18):

“An important limitation is that the timing of the boost was not randomized. This could lead to a bias in the relative efficacy of estimates at a given antibody level that compare boosted to unboosted participants (Figs. 3 and 4), although it should have minimal effect on the correlates of risk analyses that compare between antibody levels among boosted participants. The relative efficacy would be overestimated if early-boosted participants tended to be at lower risk; in contrast, the relative efficacy would be

underestimated if late/never-boosted participants had lower risk or were less likely to report COVID-19. Although we statistically adjusted for covariates to attempt to address this issue, residual confounding still remains a concern.”

3. This work measures Ab levels in arbitrary units (AU/ml) while previous work on efficacy of 2 dose vaccination used international units (IU/ml). Other studies (e.g., 34002089) used other ways to "standardize" Ab levels. Justification of the process is discussed in supplement but it is critical, and should be moved to main text. Why were the measurements done in AU and not IU advocated for in previous works on COVE trial? Could this use of different metrics skew results in some ways?

Response: The WHO has established an International Standard for anti-SARS-CoV-2 immunoglobulin that allows calibration of neutralizing antibody titers against a Spike-D614G pseudovirus to International Units/ml (for 50% inhibitory dilution titer, IU50/ml). This calibration is described in detail in the Supplementary Material and Statistical Analysis Plan provided with Gilbert et al.³ In the present manuscript, a PPD report (see p. 9 of the “Statistical Analysis Plan for Study of Post Dose 3 and Exposure-Proximal Omicron Antibody as Immune Correlates for Omicron COVID-19 in the P301 COVE Study” estimated a scaling factor of 1.04 between the PPD readout and the Duke ID50 readout. We also multiplied the PPD readout by 0.242, the conversion factor used by Duke to convert their nAb-ID50 D614G readouts to the IU50/ml scale (Supplementary Material of Gilbert et al.³). Thus, the AU/ml units for D614G nAb-ID50 titers (referred to as “ancestral strain nAbs” in the manuscript) can be transformed to International Units (IU50/ml). To clarify this, we have added the following footnote to all figures/tables that show results for ancestral strain nAbs in AU/ml: “For ancestral strain nAbs, the units AU/ml can be transformed to International Units (IU50)/ml (see SAP).” We have also added a second set of y-axis labels in Supplementary Fig. 5 with the unit label “IU50/ml” for the ancestral strain nAbs.

The conversion factor to international units was not updated for subsequent variants and thus it has not been possible to express BA.1 nAb titers in IU50/ml. However, as part of this revision, we conducted a PPD/Duke assay concordance study, for Ancestral nAbs as well as for BA.1 nAbs (see Supplementary Tables 12-14, newly added in this revision). Under the assumption that the same multiplication factor can be used to convert BA.1 nAbs in Duke-AU/ml into IU50/ml as was used for converting ancestral nAbs in Duke-IU50/ml into IU50/ml, the PPD ancestral nAbs in Duke-AU/ml were converted to IU50/ml (termed “imputed IU50/ml” in the relevant x-axis).

The x-axes of panel (a) and panel (b) in Figure 4 are now expressed in units IU50/ml and imputed IU50/ml, respectively, making the two axes directly comparable.

In the “Comparison to Ancestral Strain Correlates Study” section in Methods (pp. 25-26), we now write:

“To do this, we defined an imputed BA.1 nAb biomarker at BD29 scaled such that it can be absolutely quantitatively interpreted vs. Ancestral nAb in IU/50 ml units. This scaling

was accomplished in two steps. First, a PPD/Duke assay concordance study was performed on $n = 250$ samples (results in Supplementary Tables 12-14). The results showed that the PPD and Duke assays were highly concordant for both Ancestral and BA.1 nAb titers (Spearman correlation = 0.92 for Ancestral and 0.95 for BA.1). The concordance study also estimated Equation (5), which describes the relationship between PPD AU/ml and Duke ID50/ml (in the \log_{10} -scale) for BA.1 nAb titers:

$$(\text{PPD AU/ml} + 0.303)/1.25 = \text{Duke ID50/ml.} \quad (5)$$

Using this relationship, PPD BA.1 nAb titers in AU/ml were converted to Duke titers in ID50/ml. Second, Duke ID50/ml was converted to IU50/ml using a conversion factor of 0.242 as previously described in Gilbert et al.³ Note that the conversion factor of 0.242 for Ancestral nAbs was established based on calibration of Ancestral nAbs to the WHO anti-SARS CoV-2 Immunoglobulin International Standard (20/136). For BA.1 nAbs, given that we need to make the assumption that the same conversion factor of 0.242 (as for Ancestral nAbs) can be used for BA.1 nAbs to convert from Duke ID50/ml to IU50/ml (as the WHO International Standard for anti-SARS-CoV-2 immunoglobulin has not been assayed against Spike-BA.1 pseudovirus, to enable calibration of Duke BA.1 nAbs in ID50 to IU50/ml), we term the units of the converted BA.1 nAbs “imputed IU50/ml”. The BD29 booster relative efficacy curve analysis was repeated for this biomarker, and results overlaid with the original Day 57 vaccine efficacy curve analysis, providing a means for absolute comparison of variant-matched titer levels associated with efficacy.”

In contrast, the PPD assay readouts of anti-Ancestral (D614) Spike IgG binding antibodies in AU/ml could not be converted to international units (BAU/ml), given that no conversion factor from AU/ml to BAU/ml was developed for any of the MSD assays and that there is no equivalency study of the PPD VAC123 MSD assay compared to the VRC MSD assay that was used in the first COVE correlates study (Gilbert et al.³).

4. I found it sad that authors mention a recent study (36964146) but spend little time discussing of how the results in this paper are or are not consistent with those published previously. I also think having a more thorough discussion with results from other studies by the Davenport group could be useful, especially in settings of how Ab levels are "scaled" between different individuals and/or studies.

Response: This was an unintended omission on our part. We now provide a brief description of the results in the introduction and refer to them in the discussion.

In the introduction (p. 5) we now write:

“This question was previously investigated by Cromer et al.⁴ using a meta-analysis of 15 studies covering Ancestral, Delta, and Omicron waves and demonstrated a strong correlation between estimated nAb titer and vaccine effectiveness against COVID-19.”

Furthermore, in the discussion (p. 19) we now write:

“In Fig. 4a we showed that a post-dose 2 Ancestral nAb titer of 100 IU50/ml was associated with a 91% reduction in Ancestral COVID-19, compared to placebo, while in Fig. 4c a post-dose 3 BA.1 nAb titer of 100 imputed IU50/ml was associated with

between a 81% to 89% reduction in Omicron COVID-19 compared to an extrapolated unvaccinated control. Thus the antibody level required for approximately 90% protection is similar across variants, a conclusion aligned with the meta-analysis model of Khoury et al.,⁵ who proposed a single vaccine efficacy curve for different variants.”

5. I found the statement about data and code availability to be unacceptable.

"Access to participant-level data and supporting clinical documents with qualified external researchers may be available upon request and is subject to review once the trial is complete". First, key authors are from the NIH that requires ALL NIH-supported work to be publicly available via repositories. The statement authors provide is a double standard. Second, the data presented were clearly sufficient (per authors) for conclusions, so these anonymized data MUST be made available. Third and finally, codes for the analysis, include those with machine learning, must be made available too, for verification purposes, and as a part of reproducible science (e.g., on github).

Response: Thank you for this feedback. Both the Data Availability and Code Availability statements have been updated. First, as the trial is now completed, requests for data sharing can be considered. However while this post-hoc exploratory analysis was supported by NIH, the overall clinical trial and immunology endpoints had several sources of support. As the trial sponsor, Moderna's policy on external data sharing for completed clinical trials has been applied for this manuscript. The policy is to provide access to external researchers who provide methodologically sound proposals from 24 months after study completion, for products that have been approved by regulatory authorities. The Data Availability statement has thus been modified as follows:

Data Availability: Access to patient-level data presented in this article and supporting clinical documents with external researchers who provide methodologically sound scientific proposals will be available upon reasonable request. Such requests can be made to Moderna Inc., 200 Technology Square, Cambridge, MA 02139, email: data_sharing@modernatx.com. A materials transfer and/or data access agreement with the sponsor will be required for accessing shared data. All other relevant data are presented in the paper. The protocol is available online at ClinicalTrials.gov: NCT NCT04470427.

We have also modified our Code Availability statement as follows:

Code Availability: All analyses were done reproducibly on the basis of publicly available R scripts. A portion of these are hosted on the GitHub collaborative programming platform (https://github.com/CoVPN/correlates_reporting_moderna_booster).³² The rest of these are contained in the Supplementary Software file.

As the reviewer specifically asked about the machine learning code, we note that the SL multivariable analysis code is located in the cor_surrogates folder at the GitHub directory mentioned in the Code Availability statement (https://github.com/CoVPN/correlates_reporting_moderna_booster).

6. I like that authors list some limitations of their work. However, limitations discussed seems minor and they did not list/discuss important ones. These are critical limitations: non-randomized nature of the study, whether individuals had Omicron or not, kinetics of Ab is not fully known given few time points sampled, is HR linear as stated, etc. Limitations should focus more on the actual data and methods used and how these methods (or collected data) could be wrong or biased. Please improve.

Response: We have revised the draft as follows to discuss additional limitations of this work:

- 1) We have added a new paragraph to the discussion where we discuss the concern of the nonrandomized nature of the study (p. 18), as discussed in our response to point **(2)** above.
- 2) As to the degree of certainty of whether individuals had Omicron COVID-19 vs. COVID-19 of another lineage, in the “Omicron COVID-19 Endpoints” in Methods (p. 20) we have added: “COVID-19 cases in COVE were sequenced and we prioritized sampling cases with BA.1 lineage based on sequencing. Of the 79 naive cases, 41 were identified as BA.1 by sequencing, 26 were identified as BA.1.1 by sequencing, and 12 were inferred to be BA.1 based on COVID-19 occurring after January 15, 2022. Of the 32 non-naive cases, 16 identified as were BA.1 by sequencing, 3 were identified as BA.1.1 by sequencing, and 13 were inferred to be BA.1 based on COVID-19 occurring after January 15, 2022.”
- 3) We feel the linear kinetics after the 3rd dose are reasonable (see response to the related minor comment below).
- 4) We have investigated the linear HR using other models and found that the linear HR fits well with little improvement with alternative parametrizations (see response to the 1st major comment).

7. I find the definition of naives vs. non-naives to be a bit shaky. Given asymptomatic infections the way to determine non-naives would be to do serology with Ags that are not in the vaccine (e.g., N protein). Can you better clarify the division into naives vs. non-naives and caveats associated with such division? Perhaps showing results when you do not divide volunteers into these subgroups could be useful.

Response: The division into naives and non-naives at BD1 was based on all the information available in COVE up to BD1. Beyond COVID-19 cases, this includes anti-N serology at BD-1, at the participant decision visit (PDV), and at 6 months and 1 year post randomization. In addition, nasal swabs were conducted at the PDV. While some asymptotically infected individuals might have been missed, such individuals would have had an infection that did not result in seroconversion. Furthermore, the HR estimates of Omicron COVID-19 per 10-fold marker increase are very similar in naive and non-naive participants: 0.31 and 0.28, respectively, for BA.1 nAbs and 0.16 and

0.15, respectively, for Spike-IgG BA.1 bAbs (Fig. 2e), suggesting that naive/non-naive status does not influence the HR of Omicron COVID-19. To more formally test this, we conducted an interaction test and found no evidence that naive/non-naive status modified the HR of Omicron COVID-19. We have added the following sentence (p. 10):

“An interaction test was conducted and no evidence of naive/non-naive status modifying the HR was found (interaction p-value = 0.97 for BD29 BA.1 nAb and 0.66 for BD29 Spike IgG-BA.1 bAb).”

Moreover, we have moved the definition of naive and non-naive from the Supplementary Material to the “Trial schema and participant demographics” section of Results (p. 6), to make this definition more prominent/easy to access in the manuscript.

Minor comments

I found many figures especially in supplement being of a somewhat poor quality. One poor feature is different ranges in panels of the same graph, e.g., Figure 2 has all different scales, numbers are barely visible; same for Fig 3 and several figures in supplement. To help interpret the data, having the same range for similar measurements (e.g., Ab titer) is important.

Response: We have made several aesthetic improvements to the figures:

- Figure 2, as well as Supplementary Figs. 14-22, have all been replotted so that the x-axis range is the same for the two plots (one for naives, one for non-naives) of the same assay (e.g., nAb titer).
- Moreover, plots of the same assay and thus with harmonized x-axis ranges are now stacked vertically, to facilitate comparison.
- We have also increased the font size of axis labels, in particular those in exponent form, to enhance visibility.

I was not able to interpret 4 different styles of symbols and 2 colors in Fig 1 - perhaps use gray and red. Not sure about cases vs. non-cases division - these are hard to judge visually. Perhaps use different panels for the two groups?

Response: Thank you for this suggestion. We have simplified the key in Fig 1 to clarify that there are only two different styles of symbols: Original-vaccine arm (filled triangle) and Crossover-vaccine arm (open circle). We also have implemented the suggested color change to use gray and red instead of yellow and green for Original-vaccine arm and Crossover-vaccine arm, respectively. As the reviewer has suggested, Omicron cases (Fig. 1e, 1c, 1e, 1g) and non-cases (Fig. 1b, 1d, 1f, 1h) are separated into different panels. These stylistic changes have also all been carried over to Supplementary Fig. 5, the Ancestral equivalent of Fig. 1, as well as Supplementary Figs. 6 and 7.

Supplementary Fig. 5 - what are the red lines? Splines? It should be mentioned in the caption (same for S6). Why are there weird peaks in red lines in A and D?

Response: The formerly red lines (recolored to black in the revision) are smooth curves fitted using the LOESS method/local regression method. This information has been added to the captions of Supplementary Figs. 6 and 7: “The black lines are smooth curves delineating the relationship between the two variables and were fitted using the LOESS method/local regression method.”

A potential explanation for the small peaks in the curves in panels A and D of Supplementary Fig. 6 is that the datapoints are fairly sparse at borders of these plots, so the local regression curves can easily be biased by 1-2 datapoints there.

Supplementary Fig. 7 - move the labels from the top to the bottom. We, readers, used to look at axis names on the left and at the bottom. Same for other figures till S12.

Response: We have done this (renumbered as Supplementary Figs. 8-13 in the revision).

In main text, in addition to the 95 CIs, listing p values from the relevant test would be useful.

Response: We have added all p values in the Results sections “Correlations among antibody markers” and “Strong inverse correlations with Omicron COVID-19 risk and BD29 BA.1 markers, as well as bAb fold-rise markers, especially in naive participants”, as well as in the captions of Supplementary Figs. 10-13.

It seems that results in line 137-148 have multiple comparisons. Did you correct for multiple comparisons here?

Response: We view these correlations as descriptive statistics of marker data, so we did not do any multiple comparisons correction. Practically speaking, these correlations are all significant at $<.001$ level, so performing the correction or not would not make any impactful difference.

Describing Ab loss as a simple exponential curve may be incorrect. How different would your main results on protection be if the decay is bi-phasic as suggested in several recent studies (PMID: 37425815, 38395697) with a faster initial decay ($T_{1/2} \sim 2-3$ wks).

Response: The second reference shows kinetics over 1 year following the 3rd dose which shows exponential decay over 1 year, in contrast to primary immunization which is markedly biphasic. In our analyses the maximum follow-up from BD29 to DD1 was 106 days, suggesting the exponential decay mode is reasonable.

Using linear regression, we also tested for a relationship between the slope of decay from BD29 to DD1 and the number of days from BD29 to DD1. We reasoned that if the decay were biphasic, there might be a relationship. The p-values for D614G and BA.1 pseudovirus neutralization assay were 0.83 and 0.41; the analogous p-values for Spike IgG binding antibodies were 0.20 and 0.11.

In machine learning, how did you control for overfitting? This needs to be explained.

Response: When the ensemble machine learning methods were used to predict Omicron COVID-19 risk, their performance was measured and quantified based on the cross-validated area under the curve. Cross-validation is a model validation method that helps avoid overfitting by using out-of-sample test data.

Line 259: 95CIs include value >1 which is impossible. It seems that the authors did not do CI calculation properly. Please address.

Response: Thank you for the opportunity to clarify. A hazard ratio > 1 indicates a higher hazard rate as the ID50 titer increases. HR is not vaccine efficacy (VE) or relative vaccine efficacy (ReLVE) and need not be bounded between 0 and 1.

Line 194: how is it possible that using titers against two different strains gave the same results? Isn't that weird? This should be discussed in the Discussion. Is this due a limitation of the work?

Response: We meant to say the shape of the BA.1 and Ancestral curves are similar. The absolute level of titers for different antigens are different as is the level of protection compared to an unvaccinated control. We have now clarified this in the manuscript:

“Analyses repeated with BA.1 Spike IgG-BA.1 bAbs, Ancestral nAbs, and Spike IgG-Ancestral nAbs yielded curves with similar shape (Fig. 3b, 3d; Supplementary Fig. 33).” (bottom of p. 12)

209-212 - are these differences statistically different? This is important to discuss in light that similar vaccines do not induce high titer Abs against Omicron.

Response: Thank you for this interesting question. First, we have updated the conversion from PPD BA.1 AU/ml to IU/ml based on the new PPD/Duke assay concordance study. Please see the updated Figure 4 and the corresponding paragraphs ‘Comparison to Ancestral strain correlates study’ and ‘Using observational cohort data to attempt to infer an unvaccinated group for comparison to the boosted (three-dose) group’. The vaccine efficacy (VE) estimate of 91% at Ancestral nAb 100 AU/ml is for 2 doses of mRNA-1273 vaccine vs. placebo against Ancestral COVID-19 in the blinded phase. In contrast, the VE estimate of 71% at BA.1 nAb 100 IU/ml is for 3 doses of mRNA-1273 vaccine vs. 2 doses of mRNA-1273 vaccine (i.e. relative booster efficacy) against Omicron COVID-19 after participant unblinding. Given the many differences

between these two sets of estimates, most importantly the difference in the comparison/control group (placebo for the first two VE estimates, 2-dose vaccine for the second two VE estimates), our view is that the scientific questions are so different from each other that no formal statistical comparison can be made.

258-260 - how come naives vs. non-naives have the same HRs? Isn't that weird?

Response: We have now conducted a formal interaction test:

" An interaction test was conducted and no evidence of naïve/non-naïve status modifying the HR was found (interaction p-value = 0.97 for BD29 BA.1 nAb and 0.66 for BD29 BA.1 bAb)." (p. 10)

The interaction test is not significant, suggesting that the HR is not modified by baseline SARS-CoV-2 status. We do not have any a priori reason to believe HR would be different between two groups. One reason that the HRs were similar could be the relatively long interval between the booster and primary series.

270-272 - why is this conclusion? Some discussion is needed.

Response: In univariate analyses, p-values for analyses based on the peak titer or fold-rise markers were similar and hence the conclusion: 'The BD29/BD1 fold rise markers have similar strengths of evidence as correlates as the peak BD29 markers in univariable marker analyses.' (p. 17)

In the multivariate machine learning analysis, the predictive performance of the fold-rise markers is poorer compared to peak among naives. The predictive performance of the fold-rise markers is better than peak among non-naives. Therefore, we concluded that: 'multivariable analyses suggested weaker evidence in naive participants and stronger evidence in non-naive participants.' (p. 17)

273-280 - How do these results relate to other work published on Abs to Omicron and predicted protection? Some reflection would be important.

Response: We now compare our extrapolated CoP curve with that of the meta-analysis of Khoury et al. and conclude our results are similar to theirs. In the discussion (p. 19) we have added:

"In Fig. 4a we showed that a post-dose 2 Ancestral nAb titer of 100 IU50/ml was associated with a 91% reduction in Ancestral COVID-19, compared to placebo, while in Fig. 4c a post-dose 3 BA.1 nAb titer of 100 imputed IU50/ml was associated with between a 81% to 89% reduction in Omicron COVID-19 compared to an extrapolated unvaccinated control. Thus the antibody level required for approximately 90% protection is similar across variants, a conclusion aligned with the meta-analysis model of Khoury et al.,²⁵ who proposed a single vaccine efficacy curve for different variants."

314-316 - yes, 85-91% is lower than 96% but still surprisingly high. Why? I thought

that these vaccines are very poor protectors against COVID19 caused by Omicron.

Response: Thank you for the opportunity to clarify. We have run a new concordance study between the PPD and Duke neutralizing assays which has shifted the distribution of Omicron BA.1 IU50/ml titers. At 100 IU50/ml, the bounds on the estimated Omicron CoP curve are 81% to 89% versus 91% for Ancestral (see Figure 4c). An important point is that these estimates are for protection against an unvaccinated control; the relative efficacy against a 2-dose control is less and reflects the perception that these vaccines are poor protectors against Omicron COVID19. Indeed, at the 10th, 50th, and 90th quantiles of BD29 nAb titer the booster relative efficacies were -7%, 56% and 80%, which crudely average to less than 50%.

Reviewer #1 (Remarks on code availability):

The link was not included in the PDF of the paper, so I only saw it when I started submitting my review. I can take a look at the codes but for that I will need more time.

Response: Our updated Code Availability statement is:
All analyses were done reproducibly on the basis of publicly available R scripts. A portion of these are hosted on the GitHub collaborative programming platform (https://github.com/CoVPN/correlates_reporting_moderna_booster).⁶ The rest of these are contained in the Supplementary Software file.

Reviewer #2 (Remarks to the Author):

This is an thorough and detailed treatment of the question of protection induced by a booster dose of COVID-19 vaccine against Omicron BA1. Findings confirm that both binding and neutralising antibodies to BA1 can continue to be useful to regulators and vaccine developers as correlates of protection, with no indication that one readout is particularly to be preferred. There is therefore a likelihood that this applies to other variants as well.

Response: Thank you for the positive comments.

The paper is a bit hard to read in place - the results section has a lot of acronyms and a shorthand style of writing. This may be due to word limit and an attempt to synthesise the huge number of results presented.

Response: We have made several writing-related revisions that we feel have improved the manuscript's readability; see for example the response to point (3) from Reviewer #3, below. We have also added a Glossary of Terms Abbreviations, and Acronyms in the supplement (Supplementary Table 15); see the response to point (6) from Reviewer #3, below.

I have no suggestions for specific changes

Minor notes:

Supplementary Fig. 3 has an "XX" in it suggesting maybe something was supposed to be filled in before submission

Response: Thank you for the catch, Supplementary Fig. 3 has been updated.

Reviewer #3 (Remarks to the Author):

Zhang et al. analyze individual-level data on binding and neutralizing antibody titers (bAb and nAb, respectively) to assess the correlation with Omicron COVID-19 in a non-naïve cohort. Their main conclusion is that antibodies remain a useful surrogate endpoint for vaccine effectiveness.

The main strength of this work is the individual-level data from a large randomized controlled trial that allows Zhang et al. to address their objectives. However, the novelty and significance of their work are not made sufficiently clear, the writing lacks clarity, and it is suggested that higher antibody levels are necessary to be protected from Omicron COVID-19 compared to Ancestral COVID-19 which is not well supported by the results in this reviewer's opinion. This reviewer's recommendation is a major revision to address the major comments below.

Major comments:

1. The novelty and significance are unclear to this reviewer.

Some of the objectives of this work were already investigated using population-based data (as the authors also state in the introduction). What then is the advantage and significance of repeating this analysis using individual-level breakthrough analysis? Are their results supporting previous population-based conclusions or contradicting them?

The manuscript could be improved substantially if the authors stated explicitly the advantages of their approach.

Response: Thank you for the opportunity to clarify. The analyses presented in the article differed from the meta-analysis of Cromer et al.⁴ in several important ways. First, our analysis was an individual-level analysis within the context of a randomized trial with active surveillance. An individual level analysis differs from a meta-analysis in that it is by design harmonized in its study population (i.e. the same inclusion/exclusion criteria have been applied), laboratory assays (i.e. all nAb titers and binding antibody concentrations were measured at one laboratory), and study endpoint collection and validation (i.e. the same endpoint definition was used). In contrast, the meta-analysis of Cromer et al. included different types of study: two randomized controlled trials, seven

test-negative case-control studies, and six cohort studies. Only a subset (4) of these studies, all of which were test-negative case control (TNCC) studies, reported vaccine effectiveness estimates against Omicron COVID-19. Moreover, these studies differed in many ways, including measure(s) of vaccine effectiveness assessed (infection, PCR-confirmed symptomatic disease, hospitalization, PCR-confirmed hospital presentation, and/or PCR-confirmed hospital admission) and vaccine (mRNA vaccines, mRNA-1273, BNT162b2, ChAdOx1 nCoV-19). Another difference from our study vs. TNCC studies included in the Cromer et al. meta-analysis is that the variant-matched (i.e. Omicron) neutralizing antibody titers were not available in these studies, and thus vaccine regimen, time since vaccination, and variant of concern were used to predict neutralizing antibody titers.

Also, our analysis more directly assessed the correlation between peak and exposure-proximal antibody markers and the clinical endpoints, while a meta-analysis achieved this by pooling study-level summary statistics.

To facilitate comparison with previous work and address the issue of a 'variant-invariant' CoP curve we inferred bounds for an estimated Omicron CoP curve, with the results shown in Fig. 4c. Our results support the meta-analysis of Khoury et al., who proposed a single curve for the different variants.

2. Comparison of Ancestral correlates and Omicron correlates.

At the end of the conclusion, the authors suggest that higher titers are required for protection from Omicron COVID-19 compared to Ancestral COVID-19. This reviewer thinks that this is not well supported by the provided evidence. The proposed adjustment between Ancestral and Omicron titers is not clear (see comment below) and the authors state that it is a crude adjustment, it is not investigated whether a potential difference is significant or negligible considering vaccine efficacy (VE) confidence bounds, and the potential confounder of the variant-matching vaccine for Ancestral but not Omicron is not discussed (may there be immune priming that influences results?).

Response: We have now conducted a formal PPD/Duke assay concordance study (see Methods and Supplementary Tables 12-14) where PPD BA.1 nAb titers in AU/ml were converted to Duke ID50 and then to imputed IU50/ml in a rigorous way. The x-axes of panels a and b in Figure 4 are directly comparable, given that each is shown in either IU50/ml or imputed IU50/ml. According to the new concordance study, BA.1 nAb in the booster CoP analysis, when expressed in imputed IU50/ml unit, was half-a-log to one-log lower than the Ancestral nAb during the stage 1 analysis. Please see the updated Figure 4 and the corresponding paragraphs 'Comparison to Ancestral strain correlates study' and 'Using observational cohort data to attempt to infer an unvaccinated group for comparison to the boosted (three-dose) group.' (pp. 13-14)

In the paragraph "Comparison to Ancestral strain correlates study", the last sentences suggests that the two curves in Figure 4 are comparable but in this

reviewer's opinion they are not. The Ancestral curve shows efficacy compared to placebo (i.e., presumably relatively high risk and low efficacy) and the Omicron curve shows efficacy compared to 2 vaccine doses (i.e., presumably low risk of COVID-19 and thus high efficacy as the Ancestral vaccine provides protection also against Omicron COVID-19). For this reason, it is to be expected that the Omicron curve is below the Ancestral curve. This is addressed in the following paragraph but should also be highlighted in this section. It seems the leading sentence: 'An important question is whether a different level of variant-matched antibody is needed to 202 achieve high efficacy for Ancestral vs Omicron COVID-19' causes this reviewer some confusion.

On p. 13 we state differences between the curves in Fig. 4a and Fig. 4b pointed out by the reviewer, namely: 2-dose vs placebo compared to 3-dose vs. 2-dose, Ancestral COVID-19 vs. Omicron COVID-19, Ancestral antibody vs. BA.1 antibody:

“We next compared the Ancestral antibody/Ancestral COVID-19 VE curve (2-dose vs. placebo) estimated previously in baseline-negative participants in COVE⁴ (Fig. 4a) with the BA.1 antibody/Omicron COVID-19 booster efficacy (3-dose vs. 2-dose) curve in SARS-CoV-2 naive participants (Fig. 4b).”

These aspects are repeated in the final sentence of this section:

“Comparison of the two curves within this range of overlap shows that estimated vaccine efficacy (versus placebo) against Ancestral COVID-19 was 91% at post-dose 2 Ancestral nAb titer of 100 IU50/ml (Fig. 4a), whereas estimated vaccine efficacy (3-dose versus 2-dose) against Omicron COVID-19 was 71% at post-dose 3 BA.1 nAb titer of 100 imputed IU50/ml (Fig. 4b).”

We have removed the leading sentence that was confusing to the reviewer, given the differences between the curves shown in Fig. 4a and Fig. 4b. As we state above in the response to a minor comment from Reviewer #1, “Given the many differences between these two sets of estimates, most importantly the difference in the comparison/control group (placebo for the first two VE estimates, 2-dose vaccine for the second two VE estimates), our view is that the scientific questions are so different from each other that no formal statistical comparison can be made.”

Based on the “Using observational cohort data to attempt ...”, it is this reviewer's understanding that the VE by 13 months post dose 2 of 34% refers to protection from infection and the VE of 62% refers to protection from hospitalization, however, they appear to be used as the lower and upper bound of the VE (for protection from symptomatic infection?) at the end of this paragraph and in the conclusion (last paragraph). Please clarify the interpretation and the calculation.

In the observational study, the endpoint is either infection or hospitalization, not symptomatic COVID-19. We reasoned that the protective efficacy of 2-dose vs unvaccinated against symptomatic COVID-19 would be somewhere between the efficacy against infection (34%) and hospitalization (62%); hence, we used 34% and 62% to obtain a most conservative estimate and least conservative estimate.

The authors suggest a method to adjust the Omicron curve to compare it to the Ancestral curve (see comment above). It would be helpful to visualize this adjusted Omicron curve and directly compare it with the Ancestral curve, instead of comparing the VE at selected titers.

We have adopted your suggestion and created two curves, interpreted as the most and least conservative estimates, of 3-dose vs. unvaccinated VE against Omicron COVID-19 by BD29 BA.1 nAb titer. These are now presented in Figure 4c, facilitating direct comparison.

3. Clarity and readability.

Please give more relevant information for clarity and readability. For example, in the section “Predicted-at-exposure and BD antibody...”, the second paragraph appears to refer to naïves only – based on the caption of Figure 3 and the following paragraph – but this is not made clear in the paragraph.

Response: The section subheading contains information that the correlates of booster relative efficacy were conducted in naive participants: “Predicted-at-exposure and BD29 antibody correlates of booster relative efficacy among SARS-CoV-2 naive participants”. In the first paragraph in this section, we also state the following to reinforce this information: “We thus analyzed time-varying predicted antibody levels where the daily risk of COVID-19 depends on the predicted antibody level on that day using a Cox model with calendar time index [see Methods and ref.⁷] in naive participants.” Moreover, the final paragraph of this section explains how analogous analyses could not be conducted for non-naïves: “Analogous analyses with an unboosted control for the non-naive participants were not possible due to extreme confounding.” However, given that the reviewer still had uncertainty on the second paragraph, we have added the following (underlined): “Fig. 3a and 3b provides correlates of booster efficacy curves in naive participants for various levels of predicted-at-exposure antibody...”.

Explain terms at their first use in the text. For example, in “Antibody marker response rates and levels” paragraph 1, lines 5-6, “non-cases”, “non-naïves”, and “naïves” appear to be first used but they are not explained.

Response: We have added a new section to Results just before the “Antibody marker response rates and levels” subsection. This section, “Omicron COVID-19 study endpoint” (p. 7), provides definitions for Omicron cases and non-cases, which were previously in Supplementary Methods. We also note that “Omicron case”, “Non-Case”, and “Naïve” and “Non-naive” are all defined in footnotes of Table 1. We have also moved the definitions of SARS-CoV-2 naive and non-naive from the supplement to the first paragraph of Results, at the end of the “Trial schema and participant demographics” subsection (p. 6).

Providing motivation, introduction, or background as well as a clear theme or

topic within sections and paragraphs can improve the readability of the manuscript.

For example, no motivation, introduction, or background are included in the abstract.

Response: We have added additional background and motivation to the Abstract: “In the phase 3 Coronavirus Efficacy (COVE) trial, post-dose two Ancestral Spike-specific binding (bAb) and neutralizing (nAb) antibodies were shown to be correlates of risk (CoR) and of protection against Ancestral-lineage COVID-19 in SARS-CoV-2 naive participants. In the SARS-CoV-2 Omicron era, Omicron subvariants with varying degrees of immune escape now dominate, seropositivity rates are high, and booster doses are administered, raising questions on whether and how these developments affect the bAb and nAb correlates. To address these questions, we assessed post-boost BA.1 Spike-specific bAbs and nAbs as CoRs and as correlates of booster efficacy in COVE...”

Result section paragraphs often start with “Figure XX shows ...” and list results without providing a motivation for the analysis or connecting the different sections. Being more selective with which results are presented (and potentially making a summarizing table for other results) and then adding more explanation and a connection to presented results may also improve readability.

Response: We have made edits throughout the Results section to introduce sections with a context/motivation sentence, and to avoid beginning with wording such as “Figure XX shows...”

Examples of revised Results text include:

Section: Strong inverse correlations with Omicron COVID-19 risk and BD29 BA.1 markers, as well as bAb fold-rise markers, especially in naive participants

Instead of starting with “Supplementary Fig. 13 shows...”, we have revised to (revisions underlined): “We next assessed the BD1, BD29, and fold-rise markers as CoRs of Omicron COVID-19. First, covariate-adjusted Omicron COVID-19 risk was estimated through 92 days post-dose 3 across a range of marker levels, separately among naive and non-naive participants. As shown in Supplementary Figs. 14 and 15, no evidence of association with COVID-19 was apparent for the BD1 BA.1 or BD1 Ancestral markers, respectively.” (p. 9)

In the next paragraph, instead of starting with “Figure 2 shows....”, we have revised to (revisions underlined): “In contrast, the two BD29 BA.1 markers each inversely correlated with Omicron COVID-19 in naive participants (Figure 2a, 2b)....” (p. 10)

Two paragraphs later, instead of starting with “Supplementary Fig. 22 shows....”, we have revised to (revisions underlined): “An alternative method for assessing markers as CoRs is to divide participants into subgroups defined by antibody marker tertile (Low, Medium, High) and to compare the cumulative incidence curves and hazard ratios across the tertiles. This method differs from the previous methods in that it does not rely on any modelling assumptions. Consistent with the analyses described above, no

evidence was found to support the D1 BA.1 or Ancestral markers as inverse correlates of Omicron COVID-19, with HRs (High to Low tertile) generally close to 1 and relatively wide confidence intervals (Supplementary Table 7). In contrast, Cox-model-based marginalized COVID-19 cumulative incidence curves among naive and non-naive participants for subgroups defined by tertile of BD29 BA.1 (Supplementary Fig. 23)...” (p. 11)

Section: Predicted-at-exposure and BD29 antibody correlates of booster relative efficacy among SARS-CoV-2 naive participants

We have added the following two context/motivation sentences: “The analyses reported up to this point have considered antibody markers measured at a fixed time-point relatively close to vaccination. Given that antibody responses wane post-vaccination, however, immune responses at the time of exposure may be better correlates for COVID-19 outcomes, especially over longer follow-up periods, compared to early fixed-time-point measurements. We thus analyzed....”

Minor comments:

4. The authors group individuals as naive or non-naive and Omicron cases or non-cases and analyze, e.g., naive and non-naive individuals separately. However, in this reviewer’s opinion they do not sufficiently discuss similarities and differences between these groups. For example, GM fold-rise smaller in non-naive cases may suggest a smaller fold-change in Ab titers with increasing number of exposures (vaccine or infection). Are there other differences between these groups and what do they mean or how can they be interpreted? It would be beneficial to add a paragraph to the discussion to summarize and interpret the results in this way.

As pointed out in one of the minor comments from Reviewer #1, the HR estimates of Omicron COVID-19 per 10-fold marker increase are very similar in naive and non-naive participants: 0.31 and 0.28, respectively, for BA.1 nAbs and 0.16 and 0.15, respectively, for Spike-IgG BA.1 bAbs (Fig. 2e), suggesting that naive/non-naive status does not influence the HR of Omicron COVID-19. To more formally test this, we conducted an interaction test and found no evidence that naive/non-naive status modified the HR of Omicron COVID-19. We have added the following sentence (p. 10):

" An interaction test was conducted and no evidence of naive/non-naive status modifying the HR was found (interaction p-value = 0.97 for BD29 BA.1 nAb and 0.66 for BD29 BA.1 bAb)." (p. 10)

The interaction test is not significant, suggesting that the HR is not modified by baseline SARS-CoV-2 status.

We have added a paragraph in the discussion (p. 16):

“Given that the non-naive participants have hybrid immunity, which has been demonstrated to be quantitatively and qualitatively different from vaccination-alone

induced immunity⁷⁻¹¹, one of our objectives was to evaluate correlates separately in these two groups of participants. In our study, naive and non-naive participants were similar in age (median 52 vs. 54 years, respectively), at-risk status (27% vs. 24%, respectively), and BD29 nAbs (BA.1 nAb GMTs for non-cases: 491 AU/ml vs 572 AU/ml, respectively). Furthermore, the correlates results were very similar between the two groups, with e.g. estimated hazard ratios of Omicron COVID-19 of 0.31 and 0.28 in naive and non-naive participants, respectively, per 10-fold increase in BD29 BA.1 nAb titer. There was also no evidence of naive/non-naive status modifying the hazard ratio, as determined by a formal statistical test. Thus the naive and non-naive groups appear similar in this study.”

5. Introduction/background paragraph 2, line 4: please replace “To update our work” with a motivation or main question for this work.

Response: We have added:

“The main motivation of this article is to systematically study correlates for Omicron BA.1 and investigate to what extent nAbs and/or bAbs still have values as correlates for regulators and vaccine developers.” (p. 4)

6. In this reviewer’s opinion it would be very helpful to add a schematic (e.g., one similar to Supplementary Fig. 1 but specifically for the data used in this work) to explain the nomenclature.

Response: We have added a “Glossary of Terms, Abbreviations, and Acronyms” at the end of the Supplementary Material (Supplementary Table 15).

7. Please add p-values to the correlations in the paragraph “Correlations among antibody markers”

Response: We have now added p values to the correlation in the paragraph **Correlations among antibody markers** and also in Supplementary Figs. 10-13.

8. Conclusion, paragraph 2: the authors hypothesize that the timing of boosting affects the predictive power of baseline antibody titers. Is it possible that the number of vaccines also influences this? For example, those with a low antibody titer after 3 vaccine doses may respond less to vaccination and thus be at a higher risk of COVID-19 compared to those with a low antibody titer after 2 vaccinations (as some of them may have higher titers after the third vaccine).

Response: Thank you for this good suggestion. We have added the following (p. 17): “Another possible explanation is the difference in the number of doses (booster after 3 doses in Hertz et al. vs. after two doses in the current manuscript).”

9. Conclusion, paragraph 3: since the authors scaled the Ancestral nAb titers with a multiplicative factor to be comparable to Omicron nAb titers (in Figure 4), it is not surprising that variant-matching does not increase predictive power. The

ability to scale titers in this way could be included as further evidence to support their conclusion.

Response: Thank you for the opportunity to clarify. We agree with you that if two markers are highly correlated and concordant, then these two markers are likely to have similar predictive power. The ID50 titer against the ancestral strain and that against BA.1 had a correlation of 0.96 at BD29 (see Section **Correlations among antibody markers**). We mentioned that: ‘...the BA.1 and Ancestral markers are highly correlated...’ in paragraph 3 of the discussion section (p. 17).

References

1. Follmann D, Mateja A, Fay MP, et al. Durability of Protection Against COVID-19 Through the Delta Surge for the NVX-CoV2373 Vaccine. *Clin Infect Dis* 2024.
2. Baden LR, et al. 2024. Long-term safety and efficacy of mRNA-1273 in adults: results from open-label and booster phases of the COVE study. *Nature Communications* (In press).
3. Gilbert PB, Montefiori DC, McDermott AB, et al. Immune correlates analysis of the mRNA-1273 COVID-19 vaccine efficacy clinical trial. *Science* 2022; **375**(6576): 43-50.
4. Cromer D, Steain M, Reynaldi A, et al. Predicting vaccine effectiveness against severe COVID-19 over time and against variants: a meta-analysis. *Nat Commun* 2023; **14**(1): 1633.
5. Khoury DS, Docken SS, Subbarao K, Kent SJ, Davenport MP, Cromer D. Predicting the efficacy of variant-modified COVID-19 vaccine boosters. *Nat Med* 2023; **29**(3): 574-8.
6. CoVPN Correlates Statistics Team. Generalized Correlates Analysis Reporting. Generalized reproducible reporting workflow for statistical analyses of candidate immune correlates of risk and protection in vaccine efficacy trials for booster doses and beyond. https://github.com/CoVPN/correlates_reporting_moderna_booster Last updated 4 Dec, 2023. Access date 23 May, 2024. 2024.
7. Underwood AP, Solund C, Fernandez-Antunez C, et al. Durability and breadth of neutralisation following multiple antigen exposures to SARS-CoV-2 infection and/or COVID-19 vaccination. *EBioMedicine* 2023; **89**: 104475.
8. Bobrovitz N, Ware H, Ma X, et al. Protective effectiveness of previous SARS-CoV-2 infection and hybrid immunity against the omicron variant and severe disease: a systematic review and meta-regression. *Lancet Infect Dis* 2023; **23**(5): 556-67.
9. Sette A, Sidney J, Crotty S. T Cell Responses to SARS-CoV-2. *Annu Rev Immunol* 2023; **41**: 343-73.
10. Stamatatos L, Czartoski J, Wan YH, et al. mRNA vaccination boosts cross-variant neutralizing antibodies elicited by SARS-CoV-2 infection. *Science* 2021; **372**(6549): 1413-8.

11. Goel RR, Apostolidis SA, Painter MM, et al. Distinct antibody and memory B cell responses in SARS-CoV-2 naive and recovered individuals following mRNA vaccination. *Sci Immunol* 2021; **6**(58).

Reviewers' Comments:

Reviewer #1:

Remarks to the Author:

Additional comments

I disagree with the interpretation that Ab decay after 3rd vaccination is exponential (e.g., 38395697). It is clearly not. Plus, the data shown are not longitudinal per individual, they are for a population and thus could impact interpretation. Analysis that the authors performed to show that alternative Ab decay models do not fit the data better is not that relevant -> the data may be too sparse to tell the difference. The key question is whether assuming different models of Ab decay would give different estimates of vaccine efficacy. For example, you could assume a model of decay proposed in 38395697 or modeled after measles re-vaccination (38864816) and see how VE estimate is influenced by the model.

While authors acknowledge that their analysis and that of Khoury et al. give similar answers, they ignored the fact that the way how Ab levels are normalized to generate a single protection "curve" are different. This is important as how one scales x axis gives the answer. So, I would like to see a bit more thorough discussion of what normalization procedure is the "right" one. Obviously, following WHO metrics would be preferable but then we are dependent on few labs that are "certified" to do these measurements (Duke?) which is unfair. So, how should we measure Ab titers to predict protection from a "single" curve?

In response letter, you provide numbers for generalized Wald test. Where are p values?

Reviewer #3:

Remarks to the Author:

My previous comments on the manuscript by Zhang et al. concerned the novelty and significance, clarity, and comparison of Ancestral and Omicron correlates. The authors have sufficiently addressed all these comments (and the minor comments) and my recommendation is now to accept this manuscript (I have only very minor comments, see below).

Zhang et al. have clarified the novelty and significance, they now provide motivation for their work (e.g., in the abstract), and have made edits throughout the results section that greatly improve the manuscript (and helped this reviewer with previous misunderstandings). The authors also added more analysis that led them to revise their conclusions about the comparison of Ancestral and Omicron correlates. They now conclude that the "antibody level required for approximately 90% protection is similar across variants" (lines 414-415) – a conclusion that is supported by the provided evidence.

The main conclusion that antibody titers remains a useful surrogate endpoint and correlate with Omicron COVID-19 is supported by the authors' various and thorough analyses. This is an

important conclusion and update of previous works on correlates of protection/risk. I congratulate the authors on this work.

Minor comments:

1. Line 80: Should it be “nAb markers” instead of “Ab markers”?
2. Line 182: BD29/BD1 instead of D29/D1?
3. Lines 259 and 262: There are references here to Fig. 3c and Fig. 3d, but Fig. 3 has only 2 panels (a, b).
4. Fig. 2 a-d: Can you add the light green distribution of the marker to the figure legend?
5. Lines 503 and 509: It appears that references 25 and 26 were lost in the revision, they are not included in the references in the revised manuscript.

REVIEWERS' COMMENTS

Reviewer #1 (Remarks to the Author):

Additional comments

I disagree with the interpretation that Ab decay after 3rd vaccination is exponential (e.g., 38395697). It is clearly not. Plus, the data shown are not longitudinal per individual, they are for a population and thus could impact interpretation. Analysis that the authors performed to show that alternative Ab decay models do not fit the data better is not that relevant -> the data may be too sparse to tell the difference. The key question is whether assuming different models of Ab decay would give different estimates of vaccine efficacy. For example, you could assume a model of decay proposed in 38395697 or modeled after measles re-vaccination (38864816) and see how VE estimate is influenced by the model.

We agree that Ab decay is not exponential but biphasic as your references clearly demonstrate. The change in decay seems to happen around day 100 to 120 post peak. In our analysis, all the events occurred within 120 days of the BD29 assay date, and 44 of 47 events occurred within 80 days. Since our data is almost entirely restricted to before when the change in decay occurs, our exponential model should be accurate.

For completeness we conducted a sensitivity analysis of the exponential decay assumption by using a constant extrapolation of the predicted antibody after day 80 and where the rate of decay prior to day 80 was estimated using a linear mixed effects model. As the reviewer noted, the antibody measurements are unfortunately quite sparse and not sufficient to estimate a more flexible decay model. The estimates of the coefficient for the effect of time-varying antibody from the Cox models from the exponential decay predictions and bent-line sensitivity analysis are shown below and are very similar under both models.

BA.1 pseudoneutralization ID50 titer			
Exponential decay		Bent-line sensitivity	
Estimate (SE)	p-value	Estimate (SE)	p-value
-1.21 (0.49)	0.013	-1.23 (0.50)	0.014
BA.1 anti-Spike IgG concentration			
Exponential decay		Bent-line sensitivity	
Estimate (SE)	p-value	Estimate (SE)	p-value
-1.44 (0.67)	0.031	-1.46 (0.67)	0.029

While authors acknowledge that their analysis and that of Khoury et al. give similar answers, they ignored the fact that the way how Ab levels are normalized to generate a single protection "curve" are different. This is important as how one scales x axis gives the answer. So, I would like to see a bit more thorough discussion of what normalization procedure is the "right" one. Obviously, following WHO metrics would be preferable but then we are dependent on few labs that are "certified" to do these measurements (Duke?) which is unfair. So, how should we measure Ab titers to predict protection from a "single" curve?

Scaling by convalescent plasma is an elegant way to combine data from different studies using different assays. The protective efficacy is presented as a function of nAb titer divided by mean ancestral convalescent titer, thus putting antibody level in reference to a common standard. We used nAb titer converted to IU50/ml units based on the WHO conversion factor for ancestral nAbs. Both approaches scale the titers but use different scaling factors. The protective efficacy curve should be the same under different scaling factors though the values on the X-axis will differ. In practice the scaling factors are estimated which impacts the estimated curves. For the meta-analysis, different laboratories used different ancestral convalescent serum while our analysis required scaling from PPD to Duke and then from Duke to IU50/ml units.

We feel the 'right' approach depends on the purpose. For meta-analysis combining data from different assays/labs, convalescent scaling appears to be the only known practical way to run a combined analysis. We were fortunate to be able to run a large concordance study to scale PPD to Duke to IU50/ml and thus able to provide protection curves based on WHO standardized IU50/ml. Importantly, differential scaling does not impact the conclusion that our results align with those of Khoury et al.

We now write

“While in our study results could be interpreted in IU50/ml units based on two concordance studies (PPD to Duke for BA.1, Duke to IU50/ml for Ancestral D614G), for many correlates studies concordance testing will not be available, in which case convalescent serum scaling is the most effective and practical approach, especially when combining data from different assays and labs.” (lines 418-422)

In response letter, you provide numbers for generalized Wald test. Where are p values?

This was an oversight on our part. We now include p-values in columns to the right of the columns for Score and Generalized Wald tests.

Model	Model Equation	Degrees of freedom	Robust Score Test	Robust Score test p-value	Gen'lized Wald Test	Gen'lized Wald Test p-value
Intercept	$h_0(d) \exp\{Z[\beta_1] + X\alpha\}$,	4	19.23	0.0007	13.98	0.007
Log-Linear	$h_0(d) \exp\{Z[\beta_0 + \beta_1 Ab(d)] + X\alpha\}$,	5	32.26	<0.0001	21.94	0.0005
Quadratic	$h_0(d) \exp\{Z[\beta_0 + \beta_1 Ab(d) + \beta_2 Ab(d)^2] + X\alpha\}$,	6	32.96	<0.0001	25.50	0.0003
Spline	$h_0(d) \exp\{Z[\beta_0 + f\{Ab(d)\}] + X\alpha\}$, where $f()$ is a natural cubic with two degrees of freedom and an interior knot at the median of the distribution of $Ab(t) = 2.35$	6	33.07	<0.0001	25.21	0.0003

Response:

Reviewer #1 (Remarks on code availability):

I did not run the code but seems comprehensive (sorry, did not have the time).

Reviewer #3 (Remarks to the Author):

My previous comments on the manuscript by Zhang et al. concerned the novelty and significance, clarity, and comparison of Ancestral and Omicron correlates. The authors have sufficiently addressed all these comments (and the minor comments) and my recommendation is now to accept this manuscript (I have only very minor comments, see below).

Zhang et al. have clarified the novelty and significance, they now provide motivation for their work (e.g., in the abstract), and have made edits throughout the results section that greatly improve the manuscript (and helped this reviewer with previous misunderstandings). The authors also added more analysis that led them to revise their conclusions about the comparison of Ancestral and Omicron correlates. They now conclude that the “antibody level required for approximately 90% protection is similar across variants” (lines 414-415) – a conclusion that is supported by the provided evidence.

The main conclusion that antibody titers remains a useful surrogate endpoint and correlate with Omicron COVID-19 is supported by the authors' various and thorough analyses. This is an important conclusion and update of previous works on correlates of protection/risk. I congratulate the authors on this work.

Response: Thank you for the positive comments.

Minor comments:

1. Line 80: Should it be "nAb markers" instead of "Ab markers"?

Response: We have revised to "nAb markers measured by a pseudovirus (vs. live virus) neutralization assay" (end of the first paragraph of the Introduction).

2. Line 182: BD29/BD1 instead of D29/D1?

Response: Corrected.

3. Lines 259 and 262: There are references here to Fig. 3c and Fig. 3d, but Fig. 3 has only 2 panels (a, b).

Response: Thank you for this catch. The text should refer to Supplementary Fig. 33a and 33b. We have revised to update these references (underlined):

"Booster relative efficacy results for BD29 BA.1 nAb titer (Supplementary Fig. 33a)..." and "Analyses repeated with BD29 BA.1 Spike IgG-BA.1 bAbs (Supplementary Fig. 33b)..."

4. Fig. 2 a-d: Can you add the light green distribution of the marker to the figure legend?

Response: Added.

5. Lines 503 and 509: It appears that references 25 and 26 were lost in the revision, they are not included in the references in the revised manuscript.

Response: Thank you for this catch. We have restored these two references (now references 31 and 32 in the revision).